# The effect of seasonally and spatially varying chlorophyll on Bay of Bengal surface ocean properties and the South Asian Monsoon

Jack Giddings[1], Adrian J. Matthews[2], Nicholas P. Klingaman[3], Karen J. Heywood[1], Manoj Joshi[1], Benjamin G. M. Webber[1]

[1]Centre for Ocean and Atmospheric Sciences, School of Environmental Sciences, University of East Anglia, Norwich, NR4 7TJ, UK.
[2]Centre for Ocean and Atmospheric Sciences, School of Environmental Sciences and School of Mathematics, University of East Anglia, Norwich, NR4 7TJ, UK.
[3]National Centre for Atmospheric Science–Climate and Department of Meteorology, University of Reading, Reading, RG6 6BB, UK.

*Correspondence to:* Jack Giddings (j.giddings@uea.ac.uk)

**Abstract.** Chlorophyll absorbs solar radiation in the upper ocean, increasing mixed-layer radiative heating and sea surface temperatures (SST). Although the influence of chlorophyll distributions in the Arabian Sea on the southwest monsoon has been demonstrated, there is a current knowledge gap in how chlorophyll distributions in the Bay of Bengal influence the southwest monsoon. The solar absorption caused by chlorophyll can be parameterized as an optical parameter, $h_2$, the scale depth of absorption of blue light. Seasonally and spatially varying $h_2$ fields in the Bay of Bengal were imposed in a 30-year simulation using an atmospheric general circulation model coupled to a mixed layer thermodynamic ocean model to investigate the effect of chlorophyll distributions on regional SST, southwest monsoon circulation and precipitation. There are both direct local upper-ocean effects, through changes in solar radiation absorption and indirect remote atmospheric responses. The depth of the mixed layer relative to the perturbed solar penetration depths modulates the response of SST to chlorophyll. The largest SST response of 0.5°C to chlorophyll forcing occurs in coastal regions, where chlorophyll concentrations are high (> 1 mg m$^{-3}$), and when climatological mixed layer depths shoal during the intermonsoon periods. Precipitation increases significantly by up to 3 mm day$^{-1}$ across coastal Myanmar during the southwest monsoon onset and over northeast India and Bangladesh during the Autumn intermonsoon period, decreasing model biases.

## 1. Introduction

The strong coupling of the Indian Ocean to the atmosphere is a major factor in South Asian monsoon seasonal variability (Ju and Slingo, 1995). During the boreal summer, strong southwesterly winds transport heat and moisture from the Indian Ocean surface to sustain deep convection over the Indian subcontinent. The South Asian summer monsoon provides up to 90% of the annual rainfall for the Indian subcontinent (Vecchi and Harrison, 2002), so it is important to accurately predict the seasonal variability of monsoon rainfall given its economic importance to agriculture and other water-intensive industries.

The South Asian monsoon is initiated when lower-tropospheric winds, transporting heat and moisture, begin to flow northward from the equator to the Asian continent in response to increasing summer insolation and increasing land-sea thermal and pressure gradients (Grey arrows; Fig. 1; Webster et al., 1998). Mid-tropospheric heating from the elevated Tibetan Plateau increases the land-sea thermal and pressure contrast, further regulating the seasonal reversal of the large-scale circulation (Li and Yanai, 1996). From June to September (JJAS) high climatological precipitation rates (> 20 mm day$^{-1}$), associated with the South Asian southwest monsoon, are anchored to three locations across the Indian subcontinent: the western Ghats of southwest India, the Myanmar coast and from Bangladesh north into the Himalayan foothills (Fig. 2f–2i). Coupled atmosphere-ocean general circulation models (GCMs) have improved their representations of the seasonal variability and spatial distribution of South Asian southwest monsoon precipitation, but substantial biases remain. Lin et al. (2008) found that 12 out of 14 coupled GCMs from the Coupled Model Intercomparison Project Phase 3 (CMIP3) captured the South Asian southwest monsoon seasonal-mean precipitation rate reasonably well. However, most GCMs simulated excessive precipitation at the

Equator and insufficient precipitation across the northern BoB and Bangladesh region from May to October. Sperber et al. (2013) compared 25 CMIP5 models with 22 CMIP3 models. CMIP5 models have higher vertical and horizontal resolutions in the ocean and atmosphere and include additional earth system processes, compared with CMIP3 models. CMIP5 multi-model means have a better representation of precipitation rates over the western Ghats, Myanmar and Bangladesh than CMIP3 multi-model means from June to September. However, both the CMIP5 and CMIP3 models underestimate precipitation over the BoB and India at 20° N. There is also a consistent dry bias over central India at 25–30° N of up to 4 mm day$^{-1}$ and a delay to the summer monsoon onset and peak over most of India in both CMIP5 and CMIP3 models. The significant biases from JJAS show that state-of-the-art coupled GCMs still struggle to capture the basic seasonality of summer monsoon precipitation across the BoB and the wider Indian subcontinent.

Chlorophyll significantly affects Indian Ocean sea surface temperature (SST) and the South Asian monsoon through the absorption of sunlight (Nakamoto et al., 2000; Wetzel et al., 2006; Turner et al., 2012; Park and Kug, 2014). Nakamoto et al. (2000) used an ocean isopycnal GCM, with a two-band solar absorption scheme from Paulson and Simpson (1977), to investigate SST modulation in the Arabian Sea. Imposing a monthly climatology of chlorophyll concentrations, measured by the Coastal Zone Color Scanner (CZCS), decreased the mixed layer depth (MLD) and solar radiation penetration depth during the intermonsoon, and increased SST by 0.6°C. Wetzel et al. (2006) used a biogeochemistry model coupled to an ocean-atmosphere GCM to show that spring chlorophyll blooms in the western Arabian Sea increased SST by 1°C at 20° N that led to an increase in rainfall of 3 mm day$^{-1}$ over western India during the southwest monsoon onset. Turner et al. (2012) showed similar results when they imposed seasonally varying chlorophyll concentrations from SeaWiFS in a coupled ocean-atmosphere GCM. The spring chlorophyll blooms in the western Arabian Sea reduced MLD biases by 50%, increased SST by 0.5-1.0°C and increased rainfall by 2 mm day$^{-1}$ over southwest India during the southwest monsoon onset. Park and Kug (2014) used a biogeochemistry model coupled to an ocean GCM to investigate the biological feedback on the Indian Ocean Dipole (IOD). The response to interactive biology enhanced both warming during a positive IOD (cooling in the eastern Equatorial Indian Ocean) and cooling during a negative IOD (warming in the eastern Equatorial Indian Ocean), thus dampening the IOD magnitude, which could have significant effects on the South Asian summer monsoon.

The thermal and saline surface properties of the Bay of Bengal (BoB; Fig. 1), in the northeast Indian Ocean, are strongly forced by the monsoonal winds and large freshwater flux. In the north BoB, the large freshwater flux from river discharge and precipitation leads to strong salinity stratification and barrier-layer formation above the thermocline and below the mixed layer (Vinayachandran et al., 2002; Jana et al., 2015; Sengupta et al., 2016). The barrier layer inhibits vertical mixing (Sprintall and Tomczak, 1992; Rao and Sivakumar, 2003) and isolates the mixed layer above from cooling by entrainment (Duncan and Han, 2009), modulating the seasonal MLD and its temperature (Girishkumar et al., 2011; Shee et al., 2019).

The BoB SST rapidly responds to variations in the net surface heat flux, which in turn are primarily controlled by variations in windspeed (Duncan and Han, 2009). Although BoB SST decreases with increasing windspeed during the southwest monsoon (JJAS), SST remains high enough (> 28°C) to sustain high precipitation rates across the Indian subcontinent, consequently strengthening the salinity stratification and further reinforcing convection across the basin (Shenoi et al., 2002). The salinity stratification is weaker in the southern BoB, allowing monsoonal winds to primarily control the upper-ocean thermal structure (Narvekar and Kumar, 2006). Hence, the southern BoB MLD and SST display larger seasonal variability compared with the northern BoB (Narvekar and Kumar, 2006).

The strong BoB salinity stratification reduces biological productivity by inhibiting the vertical transport of nutrients to the sun-lit surface layers (Kumar et al., 2002; McCreary et al., 2009). Biological productivity during JJAS is also inhibited by cloud cover and by the infiltration of river sediments, which respectively reduce the incoming solar radiation at the ocean surface and the in-water penetration depth of solar radiation (Gomes et al., 2000; Kumar et al., 2010). However, in certain regions of the BoB, localised seasonal physical forcing breaks the strong stratification and increases the vertical transport of

nutrients to the sun-lit surface layers, increasing biological productivity. Chlorophyll concentrations in the coastal regions are high (> 1 mg m$^{-3}$; Fig. 1), especially near large rivers such as the Ganges, Brahmaputra, Mahanadi and Irrawaddy, because of nutrients supplied by these rivers during June–October (Amol et al., 2019). Chlorophyll concentrations in the northern coastal region typically peak in October (Fig. 3j; Lévy et al., 2007) when river discharge and nutrients also peak (Rao and Sivakumar, 2003). High chlorophyll concentrations are then transported along the northeast coast of the BoB (Amol et al., 2019).

In the southern BoB, strong southwesterly winds across the southernmost tip of India and Sri Lanka initiate coastal upwelling and thus biological productivity, leading to a maximum in chlorophyll concentration there in August (Fig. 1; Lévy et al., 2007). The Southwest Monsoon Current (SMC), a shallow, fast current, advects these high chlorophyll concentrations to the southwest BoB (Fig. 1; Vinayachandran et al., 2004). High chlorophyll concentrations are sustained east of Sri Lanka by the cyclonic (anticlockwise) eddy of the Sri Lanka Dome (SLD), where open ocean Ekman upwelling transfers nutrients to the near surface during JJAS (Fig. 1; Vinayachandran and Yamagata, 1998; Vinayachandran et al., 2004; Thushara et al., 2019). In the west and southwest BoB in winter, northeasterly winds induce open-ocean Ekman upwelling, leading to increased chlorophyll concentrations peaking in December and January (Fig. 3l–3a; Vinayachandran and Mathew, 2003; Lévy et al., 2007). Chlorophyll concentrations in the open BoB also show sub-seasonal and mesoscale variability. Surface chlorophyll concentrations are periodically enhanced by transient cold-core eddies and post-monsoon cyclones, where the strong salinity stratification is briefly eroded and nutrients are transported to the near-surface in the western and central BoB (Vinayachandran and Mathew, 2003; Kumar et al., 2007; Patra et al., 2007).

A few studies have briefly analysed the effect of seasonally varying chlorophyll concentrations on BoB upper ocean dynamics and SST, whilst also speculating how this may affect the South Asian monsoon (Murtugudde et al., 2002; Wetzel et al., 2006). Although the effect of chlorophyll on BoB SST has been demonstrated by these previous studies, the effect of chlorophyll on monsoon rainfall remains a vital knowledge gap. Without this knowledge then missing bio-physical interactions in the BoB could lead to inaccuracies in simulated air-sea interactions that are crucial in representing accurate monsoon behaviour and thus rainfall timing, location and duration over the Indian subcontinent. This study analyses the direct effect of BoB seasonally varying chlorophyll concentrations on the South Asian monsoon in an atmospheric GCM that is coupled to a mixed layer thermodynamic ocean model. A description of the experimental design, model and observed datasets used in this study is presented in Section 2. Section 3 presents the results of the control and chlorophyll-perturbed model outputs. Section 4 discusses and concludes the results from the chlorophyll-perturbed experiment.

## 2. Methods and data

### 2.1 MetUM-GOML

This study uses the Global Ocean Mixed Layer 3.0 configuration of UK Met Office Unified Model (MetUM-GOML3.0), which comprises the Multi-Column K Profile Parameterisation ocean (MC-KPP version 1.2) coupled to the MetUM Global Atmosphere 7.0 (Walters et al., 2019). The atmospheric and oceanic horizontal resolution is N216 (0.83° longitude x 0.56° latitude), which corresponds to a horizontal grid spacing of approximately 90 km. There are 85 vertical levels in the atmosphere, with approximately 50 vertical levels in the troposphere. MetUM-GOML3.0 is configured similarly to MetUM-GOML2.0 (Peatman and Klingaman, 2018) and MetUM-GOML1.0 (Hirons et al., 2015), except that the atmospheric model is updated to GA7.0 and the air-sea coupling routines are updated to couple the models via the Ocean-Atmosphere-Sea Ice-Soil (OASIS) Model Coupling Toolkit (Valcke, 2013).

MC-KPP consists of a grid of independent one-dimensional columns, with one column positioned under each atmospheric grid point at the same horizontal grid spacing as MetUM GA7.0. The ocean columns are 1000 m with 100 vertical points, with 70 points in the top 300 m; the near-surface resolution is approximately 1 m. This improves the representation of MLD and

SST, which has been shown to improve tropical convection and circulation on subseasonal scales when coupled to an atmospheric GCM (Bernie et al., 2005; Bernie et al., 2008; Klingaman et al., 2011). Each column is subject to surface forcing from freshwater, heat and momentum fluxes; vertical mixing is parameterised using the KPP scheme from Large et al. (1994). The MLD is defined as the depth where the bulk Richardson number equals the critical Richardson number of 0.3 (Large et al., 1994). The coastal region in MC-KPP is represented with columns that are partially ocean and partially land. The surface properties for ocean and land are computed separately in MC-KPP and the mean grid point temperatures are computed in the atmospheric model by combing the ocean and land surface temperatures from MC-KPP.

Solar radiation absorption is represented as a wavelength-dependent penetration depth, with blue wavelengths penetrating deeper than red wavelengths. The decay of solar irradiance through the water column is represented as a simple two-band double-exponential function (Paulson and Simpson, 1977):

$$\frac{I(z)}{I_0} = Re^{-\frac{z}{h_2}} + (1 - R)e^{-\frac{z}{h_2}} \tag{1}$$

where $I(z)$ is the solar irradiance at depth $z$; $I_0$ is the solar irradiance at the ocean surface; $R$ is the ratio of red light to the total visible spectrum; and $h_1$ and $h_2$ are the scale depths of red and blue light, respectively. The scale depth, or $e$-folding depth, is defined as the depth where solar irradiance is approximately 63% less than its surface value ($1 - e^{-1}$). Paulson and Simpson (1977) determined the optical parameters based on each of the five Jerlov water types that categorise open ocean turbidity (Jerlov, 1968). Water type IB represents the average open ocean turbidity, where chlorophyll concentrations are ~0.1 mg m$^{-3}$ (Morel, 1988); $h_1$ and $h_2$ are 1 m and 17 m, respectively. Increasing upper-ocean turbidity to water type III, where chlorophyll concentrations exceed 1.5–2.0 mg m$^{-3}$ (Morel, 1988), yields $h_1$ and $h_2$ of 1.4 m and 7.9 m, respectively. The scale depth for red light ($h_1 \sim 1 - 1.4$ m) for all water types is much less than the typical MLD (> 10 m). Hence, all red light is absorbed at the top of the mixed layer. However, the scale depth for blue light ($h_2 \sim 8 - 17$ m) is comparable to the typical MLD; a significant fraction of blue light will penetrate below the mixed layer. Hence, the reduction of $h_2$ with increasing turbidity controls the radiant heating of the mixed layer and thus SST (Zaneveld et al., 1981; Lewis et al., 1990; Morel and Antoine, 1994).

MC-KPP uses the Paulson and Simpson (1977) scheme (Eq. 1) for the absorption of red and blue light with depth through the upper ocean. In this study, $h_2$ of 17 m from Jerlov water type IB is used to represent the global average solar penetration depth in MC-KPP. Chlorophyll and biogeochemical processes are not included. The effect of chlorophyll on the ocean is modelled by specifying $h_2$.

MC-KPP does not represent horizontal or vertical advection. The ocean temperature and salinity correction method of Hirons et al. (2015) is used to constrain the MC-KPP mean state to account for missing advection and biases in atmospheric surface fluxes. The method uses a separate 10-year MetUM-GOML relaxation simulation in which temperature and salinity are relaxed with a 15-day timescale to an observed seasonal cycle, here the 1980-2009 climatology of Smith and Murphy (2007). A relaxation timescale of 15 days is optimal to produce temperature and salinity tendency terms that minimise biases in the free-running simulations (Hirons et al., 2015). A mean seasonal cycle of daily temperature and salinity tendencies that is computed from this relaxation simulation is applied to the 30-year chlorophyll perturbation simulations. The spin-up time is small (1 year) as the ocean is adjusted to a mean state after the relaxation simulation. The absence of ocean dynamics means MetUM-GOML does not represent coupled modes of variability (e.g. ENSO or IOD) that rely on a dynamical ocean (Hirons et al., 2015). The benefit of not representing these modes of variability is that the signal from the chlorophyll perturbation experiment will not be obscured by the "noise" of these interannual climate variations. The absence of full ocean dynamics also reduces computational cost and allows the model to be used for climate-length coupled simulations with shorter spin-up periods (Hirons et al., 2015).

We directly impose a seasonally varying $h_2$ value (representative of chlorophyll concentration) to selected columns within the BoB region whilst the global ocean outside the BoB region has a constant $h_2$ value. This set-up enables us to investigate the direct impact of chlorophyll on BoB surface ocean properties, atmospheric surface fluxes and the regional climate.

Furthermore, the absence of biological and physical feedbacks on chlorophyll development means that a consistent seasonally varying $h_2$ value is directly imposed on columns within the BoB throughout the simulation.


## 2.2 Chlorophyll-*a* data

To produce a temporally and spatially varying field of $h_2$ for MC-KPP, a monthly climatology of chlorophyll-*a* concentration, measured from the Moderate Resolution Imaging Spectroradiometer (MODIS) on the Aqua satellite, was used. MODIS-Aqua chlorophyll-*a* concentration (available from NASA's ocean color database; https://oceancolor.gsfc.nasa.gov) is
available as a 17-year climatology (2002–2018) at a spatial resolution of 4 km. The backscattered solar radiation from the ocean surface (water-leaving radiance) in nine spectral bands between 412–869 nm measured by MODIS-Aqua were used to calculate chlorophyll-*a* concentration (Hu et al., 2012b). Chlorophyll-*a* concentration retrievals below 0.25 mg m$^{-3}$ were calculated using the Color Index (CI) three-band reflectance algorithm (Hu et al., 2012b). Chlorophyll-*a* retrievals above 0.3 mg m$^{-3}$ were calculated using the Ocean Color 3 (OC3) algorithm, which is a fourth-degree polynomial relating three
wavelengths of water-leaving radiance (433, 490 and 550 nm) to chlorophyll-*a* concentration (O'Reilly et al., 2000). Chlorophyll-*a* retrievals from 0.25 to 0.3 mg m$^{-3}$ were calculated by merging the CI and OC3 algorithms to create the Ocean Color Index (OCI) algorithm (Wang and Son, 2016; Hu et al., 2019). Chlorophyll-*a* concentration retrievals above 5 mg m$^{-3}$ reduce the effectiveness of the OC3 algorithm (Morel et al., 2007). Organic and terrestrial material, introduced by rivers or mixed by tidal currents in coastal regions, change the scattering of visible light, affecting the water-leaving radiances (Boss et
al., 2009) and leading to an overestimate in chlorophyll-*a* concentration (Morel et al., 2007). Hence, remotely sensed chlorophyll-*a* concentrations were not determined in the eutrophic coastal regions of the Ganges and Irrawady river deltas because of the large amount of suspended organic and terrestrial material (grey shading; Fig. 1; Tilstone et al., 2011). MODIS sensor degradation on the Aqua satellite has been small (Franz et al., 2008) and all ocean color products have since been corrected and improved after cross-calibration with the SeaWiFS climatology (Meister and Franz, 2014). Chlorophyll-*a* will
henceforth be referred to as "chlorophyll" for convenience.

## 2.3 Experiment set-up

To investigate the impact of the seasonal and spatial variability of chlorophyll-induced heating in the BoB, two 30-year simulations were completed, with differing prescribed $h_2$ (representative of chlorophyll concentrations): a control run using $h_2$
= 17 m globally and a perturbation run using an annual cycle of $h_2$ at daily resolution for the BoB region (defined below) and $h_2$ = 17 m over the rest of the global ocean. In both simulations, $R$ and $h_1$ were kept constant, at 0.67 and 1.0 m respectively, representative of water type IB. The first year of both simulations was discarded due to spin up; the analysis was carried out on the remaining 29 years.

The control simulation used an effective constant global chlorophyll concentration of ~0.15 mg m$^{-3}$, which corresponds
to $h_2$ = 17 m (Jerlov water type IB; Morel, 1988). Previous studies have used control simulations with zero chlorophyll concentrations to see the full impact of chlorophyll on physical and dynamical processes (e.g. Gnanadesikan and Anderson, 2009), whilst other studies have used constant scale depths determined from parameterizations of the lowest chlorophyll concentrations encountered (e.g. Shell et al., 2003; Turner et al., 2012). Satellite observations show that the global average chlorophyll concentration for oceans deeper than 1 km is 0.19 mg m$^{-3}$ (Wang et al., 2005), similar to the value in our control
simulation.

For the perturbation simulation, the BoB region was defined as the area 77–99.5° E and 2.5–24° N (black dashed box; Fig. 3). The region extends far enough south and west to incorporate the high surface chlorophyll around the southernmost tip of India and Sri Lanka, but excludes the relatively low near-equatorial surface chlorophyll concentrations (Fig. 3f–3j). The Isthmus of Thailand and Myanmar to the east, and India and Bangladesh to the north and west, form a natural boundary to the

defined BoB region (Fig. 1). An annual cycle of daily chlorophyll concentration for MetUM-GOML was derived by linearly interpolating the monthly climatology to daily values, then regridding from the resolutions of the observations (4 km) to MetUM-GOML (~90 km).

Satellite derived chlorophyll concentrations were converted to $h_2$ using a fifth-order polynomial parameterization from Morel and Antoine (1994) (Fig. 4a–4c). This high-order polynomial relationship relates blue light from a two-band solar absorption scheme to surface chlorophyll concentrations that are assumed to have a Gaussian vertical profile in the upper ocean. The relationship shows scale depth varying as a power law function of surface chlorophyll concentration with the largest variability of scale depth (> 18 m) at the lowest concentrations (< 0.1 mg m$^{-3}$).

Missing $h_2$ data were common in regions such as the Ganges River delta due to undetermined remotely sensed chlorophyll concentrations from highly turbid coastal waters. Missing $h_2$ data in this delta extend further out onto the continental shelf during JJAS as floodwaters drain into the BoB transporting finer silt and clay further offshore (Kuehl et al., 1997). The missing $h_2$ data were typically associated with regions where the land fraction was less than 1, which includes the narrow Isthmus of Thailand and the low-lying land of the Ganges delta (black hatching; Fig. 1). A minimum of two $h_2$ values from two neighbouring data points were required to find an average $h_2$ value to fill in the missing data point. At the boundary of the BoB domain, to avoid sharp gradients the seasonally varying $h_2$ values within the BoB domain were smoothly transitioned (linearly) to the constant $h_2 = 17$ m outside the BoB domain, over a buffer region of three grid points.

Vertically integrated moisture fluxes (VIMF) were used to evaluate the water vapour transport sourced from the chlorophyll-forced BoB to the surrounding Indian subcontinent. The VIMF was calculated as

$$VIMF = \frac{1}{g} \int \vec{u} q \ dp \tag{2}$$

where $\vec{u}$ is the horizontal wind velocity, $q$ is the specific humidity, $g$ is the acceleration due to gravity, $p$ is pressure and the integration was between 1000 and 100 hPa. Note that $\vec{u}q$ was output directly by the model as monthly mean values. VIMF divergence was used to evaluate the precipitation rate changes that are due to changes in water vapour divergence. The VIMF divergence was calculated as

$$VIMFD = \frac{1}{g} \int \frac{\partial \vec{u} q}{\partial x} + \frac{\partial \vec{v} q}{\partial y} \ dp \tag{3}$$

where the integration was between 1000 and 100 hPa.

The observed monthly 18-year (1998–2015) climatological precipitation rate measured from the Tropical Rainfall Measuring Mission (TRMM) 3B42 satellite product (Huffman et al., 2007) was used to diagnose the bias in the model precipitation rate. An area-weighted re-gridding scheme was used to reduce the 0.25° horizontal resolution of the observed monthly climatological precipitation rate to match the horizontal resolution of MetUM-GOML.

## 3. Results

### 3.1 Southwest monsoon onset (April to June)

The BoB surface ocean responds to the imposed annual cycle of $h_2$ in the perturbation run during the onset of the southwest monsoon. In the central BoB, values of $h_2$ increase above the global constant of 17 m, as observed surface chlorophyll concentrations are low during southwest monsoon onset (Fig. 4a-c). Along the northern BoB coast, values of $h_2$ are as low as 5 m, as observed surface chlorophyll concentrations in coastal areas are higher than those in the central BoB (Fig. 4a-c). During May and June, the values of $h_2$ decrease and mixed-layer solar absorption increases in the northwest BoB, as observed high coastal chlorophyll concentrations extend oceanward across the continental shelf (Fig. 4b-c). In the southwest BoB, the imposed $h_2$ decreases in May and June to 14 m, as the strengthening SMC advects high chlorophyll concentrations from the south coast of India and Sri Lanka (Fig. 4b-c).

The imposed annual cycle of $h_2$ directly affects coastal SST. During April the increase in solar absorption by chlorophyll along the northern and western coastal regions significantly (at 5% level) increases monthly average SST by 0.5°C (Fig. 4d). Correspondingly, the monthly average 1.5 m air temperature increases by 0.5°C in the perturbation run (Fig. 4g). The strengthening alongshore wind over the warmer western coast results in a large increase in upward latent heat flux of 20 W m$^{-2}$ (Fig. 4j). This increase in atmospheric moisture leads to an anomaly in the VIMF of 30 kg m$^{-1}$ s$^{-1}$ (Fig. 5a) that is in the same direction as the mean VIMF in the control run (Fig. 6a). The increase in VIMF converges over northeast India and Bangladesh as shown by the negative VIMF divergence (Fig. 5a), supplying extra moisture needed for the increase in precipitation rate of 2 mm day$^{-1}$ (significant at the 5% level; Fig. 4m).

The increase in solar absorption in the mixed layer by high chlorophyll concentrations persists during May and June along the coasts (Fig. 4b and 4c). Low $h_2$ along the northern and western BoB coastal regions acts to increase monthly average SST by 0.5°C (Fig. 4e and 4f). The upward latent heat flux increases (Fig. 4k and 4l) due to an increase in the specific humidity at the surface, which is associated with the higher SST. This increase in SST is therefore offset by the negative feedback from the latent heat flux increase.

In June, the precipitation rate over the Myanmar coast increases by 3 mm day$^{-1}$ (significant at the 5% level; Fig. 4o). The monthly average 1.5 m air temperature increases by 0.4°C (Fig. 4i), which is caused by an increase in SST (Fig. 4f) where $h_2$ along the western BoB is low (Fig. 4c). The upward latent heat flux increases by 10 W m$^{-2}$ (Fig. 4l) and the VIMF increases by 20 kg m$^{-1}$ s$^{-1}$ (Fig. 5b) in addition to a strengthening southwesterly moisture transport during the southwest monsoon onset (Fig. 6b). The enhanced convergence of VIMF over the Myanmar coast (Fig. 5b) supplies the moisture for the increase in precipitation rate (Fig. 4o). We have thus demonstrated that high coastal chlorophyll concentrations perturb the absorption of solar radiation that increases air temperature and SST, which significantly increases spring intermonsoon precipitation rates in the northern and eastern BoB.

### 3.2 Southwest monsoon (July to October)

The values of $h_2$ continue to decrease in the BoB open ocean into July and August (Fig. 7a and 7b), as high chlorophyll concentrations off the continental shelf and SMC encroach further into the central BoB. The lowest monthly average $h_2$ in the SMC region and the central BoB occurs in August with a value of 12 m and 15 m, respectively (Fig. 7b). During September and October the SMC weakens and the observed high chlorophyll concentrations retreat back to the coast, increasing average $h_2$ in the SMC region and the central BoB to around 16 m (Fig. 7c and 7d). Meanwhile, along the northwest BoB during October, monthly average $h_2$ decreases to 13 m, as observed high chlorophyll concentrations retreat back onto the continental shelf (Fig. 7d).

BoB surface ocean and regional climate respond to the above changes in $h_2$ during JJAS. Higher coastal SSTs (significant at the 10% level) are co-located with the high coastal chlorophyll concentrations, whereas open-ocean SST is largely unchanged by BoB chlorophyll forcing (Fig. 7e–7g). In July, an increase in SST and a slight increase in alongshore windspeed over the west BoB increases the upward latent heat flux (Fig. 7m), but this does not significantly change precipitation rate (Fig. 7q). In August, a further increase in the alongshore windspeed increases the magnitude and spatial extent of the upward latent heat flux across the northern BoB (Fig. 7n). During September an increase in windspeeds over the northern Myanmar coast increases surface ocean evaporation (Fig. 7o). The VIMF increases in magnitude and remains approximately in the same direction as the mean VIMF in the control run (Fig. 5c and 6c). Negative VIMF divergence over the northern Myanmar and Bangladeshi coast in the perturbation run (Fig. 5c) supplies moisture for the increase in precipitation rate in this region (significant at the 5% level; Fig. 7s).

By October the combined atmospheric moisture sourced from the warmer western BoB and Andaman Sea leads to an increase in precipitation rate of up to 3 mm day$^{-1}$ over west Bangladesh and northeast India (significant at the 5% level;

Fig. 7t). The spatial extent of the increased precipitation rate is considerably larger than previous months, extending further west over the Indo-Gangetic plain and encompassing megacities such as Kolkata and Dhaka. An area-weighted 29-year monthly average precipitation rate over west Bangladesh and northeast India (20–25° N, 85–90° E; black dashed box in Fig. 7t) shows a rainfall maximum in August in both simulations (Fig. 8a). The precipitation rate differences gradually increase from July to August and peak in October at 2 mm day$^{-1}$ (Fig. 8b). Alongshore winds over the warmer Isthmus of Thailand and the coast of Myanmar further increase atmospheric moisture transport to the northern BoB (Fig. 6d). The upward latent heat flux increases by 13 W m$^{-2}$ (Fig. 7p) and the VIMF increases by 30 kg m$^{-1}$ s$^{-1}$ over the coast of Myanmar (Fig. 5d). The negative VIMF divergence over west Bangladesh and northeast India supplies moisture for the increase in precipitation rates in this region (Fig. 5d). As in the spring intermonsoon, the increase in precipitation rate during autumn intermonsoon in the northern BoB is primarily attributed to high coastal chlorophyll concentrations and increased SST extending from the Andaman Sea to the Ganges river delta along the chlorophyll-perturbed BoB coastal region.

The enhanced convective activity over west Bangladesh and northeast India during October is associated with an increase the vertical wind velocity at the 500 hPa pressure level (Fig. 9a). At the 200 hPa pressure level enhanced westerly winds converge over eastern China (Fig. 9b), which leads to increased subsidence (Fig. 9a). This subsidence reduces precipitation and increases surface temperature over eastern China (significant at the 5% level; Fig. 9c and 9d). This indirect remote response resembles the effect of the "Silk Road" pattern; a stationary Eurasian-Pacific Rossby wave train that occurs during the Northern Hemisphere summer and produces significant air temperature and rainfall anomalies in east Asia (Ding and Wang, 2005).

### 3.3 Mixed layer radiant heating and SST modulation

The hypothesised direct link between a change in $h_2$ and a resultant change in SST is examined in more detail in this subsection. The radiant heating rate of the mixed layer, and resultant change in SST, depends not only on $h_2$, but also on changes in the surface flux of shortwave radiation, which is dependent on cloud cover, and changes in the depth of the mixed layer. Here, we assess which of these three factors is primarily responsible for the changes in the radiant heating rate of the mixed layer.

We assume that the red-light radiative flux is absorbed within approximately the top 1 m and entirely within the mixed layer, and only the blue-light radiative flux can partially penetrate below the mixed layer. The radiant heating rate of the mixed layer is calculated as

$$\text{RHR} = \left.\frac{dT}{dt}\right|_Q = \frac{Q_{sw}(0) - (1-R)Q_{sw}(0)e^{-\frac{H}{h_2}}}{\rho c_p H} \tag{4}$$

where $T$ is the temperature of the mixed layer; $t$ is time; $Q_{sw}(0)$ is the monthly 29-year average downward shortwave radiation flux incident at the ocean surface; $\rho = 1025$ kg m$^{-3}$ is the density of the mixed layer; $c_p = 3850$ J kg$^{-1}$ K$^{-1}$ is the specific heat capacity of sea water; $R = 0.67$ is the ratio of red light to total visible light for Jerlov water type IB; H is the monthly 29-year average MLD; and $h_2$ is the monthly average $h_2$ that was imposed in the control and perturbation run.

Within the BoB, the largest imposed change in $h_2$ is 13 m. Assuming that the other variables remain constant, a change in $h_2$ of 13 m changes radiant heating rates by 0.4°C month$^{-1}$. The largest model change in downward shortwave radiation is 14 W m$^{-2}$, which changes radiant heating rates by 0.3°C month$^{-1}$, comparable to the change from $h_2$ variations. The largest model MLD change is 3 m, which changes radiant heating rates by 0.4°C month$^{-1}$, also comparable to the change from $h_2$ variations.

We compare the mixed layer radiant heating rates of the control and perturbation runs during June and October (Fig. 10a and 10b). We focus on two regions: the open ocean region of the SMC (83–86° E, 5–8° N; black boxes in Fig. 10) and the coastal region of the Irrawaddy Delta (95–98° E, 14–17° N; black boxes in Fig. 10). The two regions are an important source

of heat and moisture for the June and October precipitation rate perturbations and display distinctive chlorophyll regimes. The SMC is an open ocean region that displays large seasonal changes in $h_2$, whilst the Irrawaddy Delta is a coastal region that displays continuously low $h_2$. In June and October, coastal regions have the highest radiant heating rate difference between the control and perturbation runs (Fig. 10a and 10b). In June, the area-weighted mean radiant heating rate in the coastal region of the Irrawaddy Delta increases by 0.4°C month$^{-1}$ in the perturbation run (Fig. 10a). An $h_2$ decrease of 9 m has the largest contribution to the radiant heating rate increase of 0.7°C month$^{-1}$ (Fig. 11a), compared with an MLD decrease of 0.2 m (Fig. 10e), which contributes to less than 0.1°C month$^{-1}$ (Fig. 11e). A decrease in downward shortwave radiation flux of 8 W m$^{-2}$ (Fig. 10c), associated with an increase in monsoon cloud cover, cools the region by 0.3°C month$^{-1}$ (Fig. 11c). In October, the radiant heating rate difference in the Irrawaddy Delta increases by 1.5°C month$^{-1}$ in the perturbation run (Fig. 10b). The radiant heating rate difference is larger than June because of an increase in monthly average downward shortwave radiation flux and a shallower MLD in both the control and perturbation runs. A decrease in $h_2$ of 9 m has the largest contribution to the radiant heating rate increase of 1.4°C month$^{-1}$ (Fig. 11a), whereas, a decrease in the MLD of 0.1 m (Fig. 10f) and an increase in downward shortwave radiation flux of 1 W m$^{-2}$ (Fig. 10d) only contribute to less than 0.1°C month$^{-1}$ of the increase in radiant heating rate respectively (Fig. 11d and 11f). The changes in $h_2$ are more influential on mixed layer radiant heating rates and SSTs compared with small changes in MLD and downward shortwave radiation flux in the Irrawaddy Delta during June and October.

In June, the area-weighted mean radiant heating rate difference in the SMC region decreases by 0.1°C month$^{-1}$ in the perturbation run (Fig. 10a). A decrease in the downward shortwave radiation flux of 5 W m$^{-2}$ (Fig. 10c) has the largest contribution to the radiant heating rate decrease of 0.1°C month$^{-1}$ (Fig. 11c), whereas, a decrease in $h_2$ of 2 m and an increase in MLD of 0.4 m (Fig. 10e) contribute less than 0.1°C month$^{-1}$ to the radiant heating rate (Fig. 11a and 11e). In October, the radiant heating rate difference of the SMC region shows an increase of 0.1°C month$^{-1}$ (Fig. 10b). A decrease in $h_2$ of 3 m has the largest contribution to the radiant heating rate increase of 0.1°C month$^{-1}$ (Fig. 11b), whereas, a decrease in downward shortwave radiation flux of 1 W m$^{-2}$ (Fig. 10d) and an increase in MLD of 0.2 m (Fig. 10f) contribute less than 0.1°C month$^{-1}$ to the radiant heating rate (Fig. 11d and 11f). In the SMC region, changes in $h_2$ are smaller than those in coastal regions during June and October. Thus, changes in $h_2$ and indirect changes in MLD and downward shortwave radiation exert a comparable control on open ocean mixed layer radiant heating rate and SST.

The radiant heating rate of the mixed layer, and resultant change in SST, further depends on the seasonal changes to the depth of the mixed layer relative to the solar penetration depth (Turner et al., 2012). Here, we examine how the depth of the mixed layer relative to the solar penetration depth affects mixed layer radiant heating rates and SSTs for the open ocean region of the SMC and the coastal region of the Irrawaddy Delta during June and October.

In the Irrawaddy Delta region during October, the MLD shoals to 9 m (green dashed line; Fig. 12b), which is similar to the perturbed $h_2$ (green dot; Fig. 12b). When the mixed layer is shallow, the increased near-surface radiant heating from reducing $h_2$ is distributed to a shallower depth, increasing the average change in the radiant heating rate by 1.2°C month$^{-1}$ ($\Delta dT/dt$; Fig. 12f). Below 10 m depth radiant heating rates reduce due to reduced $h_2$. There is also no change in MLD in response to reduced $h_2$ in the perturbation run. The increase in local wind speed of 0.8 m s$^{-1}$ is likely to have de-stratifying effects on the upper ocean that oppose the stratifying effects of increased mixed layer radiant heating. When the MLD deepens below 10 m, the biological-induced effects of the increased radiant heating rates above 10 m and reduced radiant heating rates below 10 m are mixed, reducing the net effect of biological heating on mixed layer temperature. In June, the MLD deepens to 16 m (Fig. 12a), resulting in a smaller average radiant heating rate change of 0.4°C month$^{-1}$ (Fig. 12e). Consequently, the October SST increases by 0.5°C, compared with a smaller increase of 0.2°C in June. Hence, shoaling the mixed layer to a

depth comparable to the perturbed solar penetration depth in October limits the turbulent mixing processes to a depth where chlorophyll perturbs solar radiation absorption, and makes SST more sensitive to chlorophyll concentration changes.

In the SMC region during October, the MLD shoals to 28 m (Fig. 12d), approximately twice the depth of the perturbed $h_2$, resulting in an average change in the mixed layer radiant heating rate of 0.1°C month$^{-1}$ (Fig. 12h). As in the Irrawaddy Delta region, there is no change in MLD in response to biological warming in the SMC region due to an increase in local wind speed of 0.8 m s$^{-1}$, which is likely to oppose the stratifying effects of increased mixed layer radiant heating. During June, the MLD extends to 36 m (Fig. 12c), resulting in an average change in the mixed layer radiant heating rate below 0.1°C month$^{-1}$ (Fig. 12g). As in the Irrawaddy Delta region, the effect of chlorophyll on upper ocean temperature depends on the MLD in the SMC region, with the shallowest MLD and largest change in radiant heating rate in October. With lower chlorophyll concentrations in the SMC region than the Irrawaddy Delta region, the resultant change in SMC regional average radiant heating rate in the top 10 m is considerably lower.

## 4. Discussion and Conclusions

In this study, we have identified that the influence of biological warming on the South Asian monsoon strongly depends upon the seasonality of the chlorophyll concentration and the depth of the mixed layer, which is further dependent on the timing of the monsoon itself. The effect of chlorophyll on SST is amplified during the intermonsoon periods when shallow MLDs are comparable to the perturbed solar penetration depths. The MLD, and its effect on the biological warming, varies seasonally and spatially in the BoB. Coastal regions experience larger SST increases than open ocean regions because of higher chlorophyll concentrations and shallower MLDs. The SST increase is larger during the autumn intermonsoon (September–October) than the spring intermonsoon (April–May) and southwest monsoon onset (June). During the spring intermonsoon, chlorophyll concentrations are low across the open ocean, but remain high in coastal regions. During the southwest monsoon onset chlorophyll concentrations are high when the MLD is relatively shallow (< 30 m) in the northern and western coastal BoB, leading to increased SST. During the autumn intermonsoon, high chlorophyll concentrations extend over the continental shelf in the northern BoB, the SMC region and the eastern BoB, in contrast to the spring intermonsoon where high chlorophyll concentrations are confined to the coasts. The chlorophyll concentrations in the southwest and northwest BoB peak in August and October respectively (Lévy et al., 2007), whilst the MLD is shallowest across the basin, which results in an increase in mixed layer radiant heating rate and SST in the western BoB in autumn.

The direct changes in $h_2$ in coastal regions are large, and thus more influential on mixed layer radiant heating rate and SST. The resultant increase in the radiant heating rate of the coastal mixed layer and SST during the southwest monsoon onset and autumn intermonsoon increases the latent heat flux and transport of moisture to the Indian subcontinent. Precipitation rates over the Myanmar coast during the southwest monsoon onset increase by 3 mm day$^{-1}$. Comparing the monthly average precipitation rate difference (Fig. 4o) with the control simulation bias (Fig. 13a) shows that the model dry bias of 4 mm day$^{-1}$ over the Myanmar coast is partly removed in the perturbation run. Precipitation rates over western Bangladesh and northeastern India during the autumn intermonsoon increase by 3 mm day$^{-1}$. Comparing the precipitation differences (Fig. 7t) with the model bias (Fig. 13b) shows that the model dry bias of up to 3 mm day$^{-1}$ over northeast India is removed in the perturbation run. The reduced model biases after imposing a more accurate representation of chlorophyll further highlights the importance of including chlorophyll in coupled models.

Fig. 14 illustrates and summarises the effect of chlorophyll during the summer monsoon onset in the BoB interior where mixed layers are shallow (30 m) and where there is a zonal gradient in mixed layer turbidity. Chlorophyll concentrations are high (1 mg m$^{-3}$) and values of $h_2$ are low (8 m) to the west, and chlorophyll concentrations are low (0.1 mg m$^{-3}$) and values of $h_2$ are high (20 m) to the east. Mixed layer radiant heating rates would increase ($\Delta$RHR > 0) in the high chlorophyll

concentration region, which reduces radiant heating rates ($\Delta$RHR < 0) below the mixed layer. Increasing the mixed layer radiant heating rate increases mixed layer temperature and SST ($\Delta$SST > 0). The upward latent heat flux and evaporation increases with increasing SST and strengthening monsoon winds. Convergence of the additional lower-tropospheric moisture that is transported by the monsoon winds increases the precipitation rates to the east.

During October, the enhanced precipitation rate and convective activity in the northern BoB perturbs upper-tropospheric winds, potentially causing reduced precipitation rates over eastern China, similar to the Silk Road effect. The Silk Road pattern has been found to influence extreme heat waves over eastern China, causing considerable socio-economic devastation (Thompson et al., 2019). Indeed, the model does display significantly warmer surface temperatures in this region at this time (Fig. 9c). The Silk Road pattern dynamics have been previously linked to the South Asian summer monsoon (Stephan et al., 2019). Diverging upper-tropospheric winds caused by precipitation anomalies over the Indian subcontinent interact with midlatitude westerlies, which influences the strength and positioning of the subtropical northwestern Pacific anticyclone over eastern China (Ding and Wang, 2005; Hu et al., 2012a). The effect of chlorophyll on the midlatitude Rossby wave train and its potential impact on East Asian climate needs further investigation.

Turner et al. (2012) identified a similar modulation of the seasonal SST cycle by MLD after imposing seasonally varying chlorophyll concentrations in the Arabian Sea. During the spring intermonsoon, a peak in surface chlorophyll concentrations and shallow MLDs led to an increase in SST. During the autumn intermonsoon, another peak in surface chlorophyll concentration led to a similar, but weaker increase in SST due to deeper MLDs and stronger turbulent surface fluxes. The BoB has less biological productivity than the Arabian Sea because of light and nutrient limitation (Kumar et al., 2002), though chlorophyll concentrations in the coastal BoB can be as high as in the Arabian Sea. The BoB is also exposed to the same monsoonal winds as the Arabian Sea. Such localised, physical forcing modulates the MLD, which in turn modulates the biological warming. Hence, the SST increase of 0.5°C in coastal regions of the BoB during the spring and autumn intermonsoons is similar to the increase in SST in the Arabian Sea during the spring intermonsoon.

Previous studies show that the effect of biological warming is amplified due to secondary feedbacks on MLD. In the Arabian Sea, high chlorophyll concentrations increase solar radiation absorption and so increase thermal stratification, which inhibits vertical mixing, shoals the MLD and further increases SST (Nakamoto et al., 2000; Wetzel et al., 2006; Turner et al., 2012). In our study, secondary feedbacks on the MLD are consistent in magnitude with the Arabian Sea studies. The maximum MLD difference is 3 m in the central BoB during June. Coastal MLDs shoaled around the southernmost tip of India and the northern BoB in June by ~1 m and MLDs shoaled around the Isthumus of Thailand in October by ~1 m (Fig. 10e and 10f). The effect of high chlorophyll concentrations in these localised coastal regions appears to have altered upper-ocean thermal stratification when there is little or no change in windspeed, while in the majority of the BoB, changes in windspeed primarily alter upper-ocean thermal stratification.

In our study, a realistic chlorophyll distribution increased open ocean SST by ~0.1°C and increased coastal SST by ~0.5°C during the intermonsoons and southwest monsoon onset. The simulated increase in open ocean SST is consistent with previous work (Murtugudde et al., 2002; Wetzel et al., 2006). However, the increase in coastal SST, primarily in the eastern BoB coastal region, is larger in magnitude than previous work: Wetzel et al. (2006) underestimated seasonal chlorophyll concentrations in the BoB coastal regions, while Murtugudde et al. (2002) used a low-resolution annual mean chlorophyll concentration which removed the seasonal variability of chlorophyll concentration, whereas we impose an annual cycle of daily $h_2$ across the BoB. Hence, the coastal and open ocean SST responses are more accurately represented here than in previous work.

The derivation of the imposed annual cycle of $h_2$ in coastal regions has limitations. Firstly, the ocean colour algorithms used to determine chlorophyll concentrations from satellite are not completely effective in turbid coastal waters (Morel et al., 2007; Tilstone et al., 2013). Organic and inorganic constituents such as Coloured Dissolved Organic Matter (CDOM) and suspended sediments strongly attenuate blue light and are thus falsely identified as a chlorophyll-*a* pigment, which typically

leads to an overestimation in chlorophyll concentration (Morel et al., 2007). Secondly, the Morel and Antione (1994) chlorophyll parameterisation is not applicable for coastal waters, as the parameterisation is based on the absorption by chlorophyll-*a* pigments and not by the attenuation of other in-water constituents. Missing $h_2$ values in the Ganges river delta are interpolated from neighbouring $h_2$ values that are likely associated with satellite product and parameterisation uncertainty. The Ganges coastal region has been found to influence spring intermonsoon SST and precipitation rates in the northern BoB. Possible positive biases in chlorophyll concentration in the Ganges river delta are likely to lead to an overestimation in the coastal biological warming, SST and precipitation rate increase. Ocean colour algorithms to determine proxy coastal chlorophyll concentrations are still an area of active research (Blondeau-Patissier et al., 2014). Future studies should consider the attenuation of solar radiation from other oceanic constituents in turbid coastal regions to better represent radiant heating in the upper ocean.

CDOM is a common oceanic constituent that perturbs solar penetration depths. The derived values of $h_2$ incorporate the bio-optical property of chlorophyll-*a* pigment concentration, largely excluding CDOM. CDOM increases the radiant heating rate of nearshore coastal waters of North America (Chang and Dickey, 2004) and in the Arctic (Hill, 2008). Imposing an annual mean of remotely sensed CDOM absorption coefficients in a coupled ocean-atmosphere GCM reduced solar penetration depths and increased coastal SST in the Northern Hemisphere during the summer (Kim et al., 2018). CDOM concentrations are high in the western and northern coastal regions of the BoB at the mouths of major rivers (Pandi et al., 2014). Thus, including the bio-optical properties of CDOM and other biological constituents would likely increase coastal SST in the BoB, with additional implications for regional climate.

The chlorophyll concentration in the BoB upper ocean is not homogeneous with depth. In situ observations show that the vertical depth of chlorophyll maxima varies between 10 and 80 m (Thushara et al., 2019; Pramanik et al., 2020), often occurring at depths undetected by satellite radiometer sensors (Huisman et al., 2006). Variations in the vertical depth of the chlorophyll maxima would vary the vertical depth of enhanced radiant heating. However, if the depth of the chlorophyll maxima occurs at a depth where solar radiation is significantly reduced (e.g., at the euphotic depth where solar radiation is ~1% of its surface value), then the change in local radiant heating at that depth would be negligible (Morel and Antione, 1994). Indeed, observations show the occurrence of intense deep chlorophyll maxima in the BoB at depths of 20 to 40 m (Thushara et al., 2019), which might have a strong influence on local mixed layer radiant heating and vertical heat distributions. Hence, the effect of nonuniform chlorophyll concentration profiles on upper ocean radiant heating and SST requires further investigation.

The mesoscale and sub-mesoscale spatial variability of $h_2$ and associated oceanic processes is inadequately represented in MC-KPP due to its coarse horizontal resolution. The coastal region in MC-KPP is represented by multiple grid points that are partially ocean and partially land at an approximate 90 km horizontal resolution. Such a resolution means that at the coastlines, the mesoscale coastal chlorophyll concentration features and the corresponding solar penetration depths are poorly resolved. Future studies should consider using a high-resolution, fully dynamical model to accurately resolve the coastline and associated solar penetration depths. The simulated dynamics would improve the representation of mesoscale eddy activity along the coast and open ocean, which increases biological productivity (Kumar et al., 2007) that in turn increases local solar radiation absorption.

The sub-seasonal temporal variability of $h_2$ is inadequately represented in MC-KPP due to the use of a monthly mean climatological chlorophyll concentration at a reduced horizontal resolution. In reality, the advection of high surface chlorophyll concentrations into the south and central BoB varies with the strength and positioning of the SLD and SMC (Vinayachandran et al., 2004), which is itself further influenced by local wind stress and seasonal Rossby waves (Webber et al., 2018). Surface chlorophyll concentrations are periodically enhanced by transient cold-core eddies and postmonsoon cyclones in the central BoB, which briefly upwell nutrients to the ocean surface (Vinayachandran and Mathew, 2003; Patra et al., 2007). In coastal regions, nutrient concentrations, which affect surface chlorophyll concentrations, vary with river discharge (Kumar et al.,

2010). Suspended terrestrial sediment that perturbs solar penetration depths on the continental shelf also depend on river discharge (Kumar et al., 2010; Lotliker et al., 2016). All these factors influence solar penetration depths on timescales of days to weeks and on spatial scales of less than 1 km. By smoothing over the large subseasonal variability of chlorophyll concentration, such variations in solar penetration depth are not represented in the present study.

The limitations of representing ocean dynamics as a mean seasonal cycle means that MC-KPP cannot capture any ocean dynamical response to biologically-induced changes to ocean properties (e.g., changes to ocean temperature and salinity transports). Previous studies have shown large effects of chlorophyll on ocean dynamics in the equatorial Pacific (e.g., Nakamoto et al., 2001; Murtugudde et al., 2002) and in mid- to high-latitude regions (e.g., Manizza et al., 2005; Patara et al., 2012). Modified biological warming at the surface or perhaps modified solar radiation penetration below the mixed layer could

affect the dynamics of SMC and SLD in the BoB. Missing modes of variability in MetUM-GOML that rely on a dynamical ocean, such as ENSO and IOD, could combine non-linearly with the ocean anomalies induced by biological warming, with implications for monsoon rainfall. Further research using a fully dynamical coupled ocean-atmosphere GCM is required to show the dynamical changes and feedbacks of biological warming on the BoB oceanic and atmospheric system.

    Biological heating has complex physical and dynamical feedbacks in the ocean, which in turn imply similar feedbacks on

BoB biological processes. The imposed seasonally and spatially varying $h_2$ in MC-KPP eliminates any biological response to secondary feedbacks in the ocean. A coupled biogeochemistry model linked to an ocean-atmosphere GCM would be needed to further understand secondary feedbacks on phytoplankton productivity. Secondary feedbacks may include changes to cloud cover that affect the incoming shortwave radiation needed for biological productivity; changes to thermal and salinity stratification that affect the vertical mixing of nutrients to the ocean surface; or changes to rainfall that affect river discharge

and nutrient availability on the continental shelf that influence biological productivity. The resultant changes to biological productivity could either enhance or deplete chlorophyll concentrations at the surface, with further implications to the spatial and temporal extent of biological heating. It is important that realistic simulations of chlorophyll concentrations are included as an additional Earth system process in high-resolution coupled ocean-atmosphere GCMs, which may improve the simulated seasonality and intraseasonal variability of the South Asian monsoon.


**Acknowledgements**

The Bay of Bengal Boundary Layer Experiment (BoBBLE) project is a joint program funded by the Ministry of Earth Sciences,

Government of India, and NERC in the United Kingdom. JG PhD project was supported by the NERC EnvEast DTP (NE/L002582/1). AJM, KJH, BGMW and MJ were supported by NERC (NE/L013827/1) and NPK was supported by a NERC Independent Research Fellowship (NE/L010976/1). The National Centre for Atmospheric Science supported the assembling of MetUM-GOML3.0 and development of MC-KPP. The MODIS-Aqua monthly climatological chlorophyll-*a* concentration product, used for the perturbation simulation, is available from NASA's ocean color database

(https://oceancolor.gsfc.nasa.gov). The monthly climatological precipitation rate 3B42 product measured by the Tropical Rainfall Measuring Mission (TRMM) satellite is available from NASA's Goddard Earth Sciences Data and Information Services Centre (GES DISC; http://daac.gsfc.nasa.gov/precipitation).

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

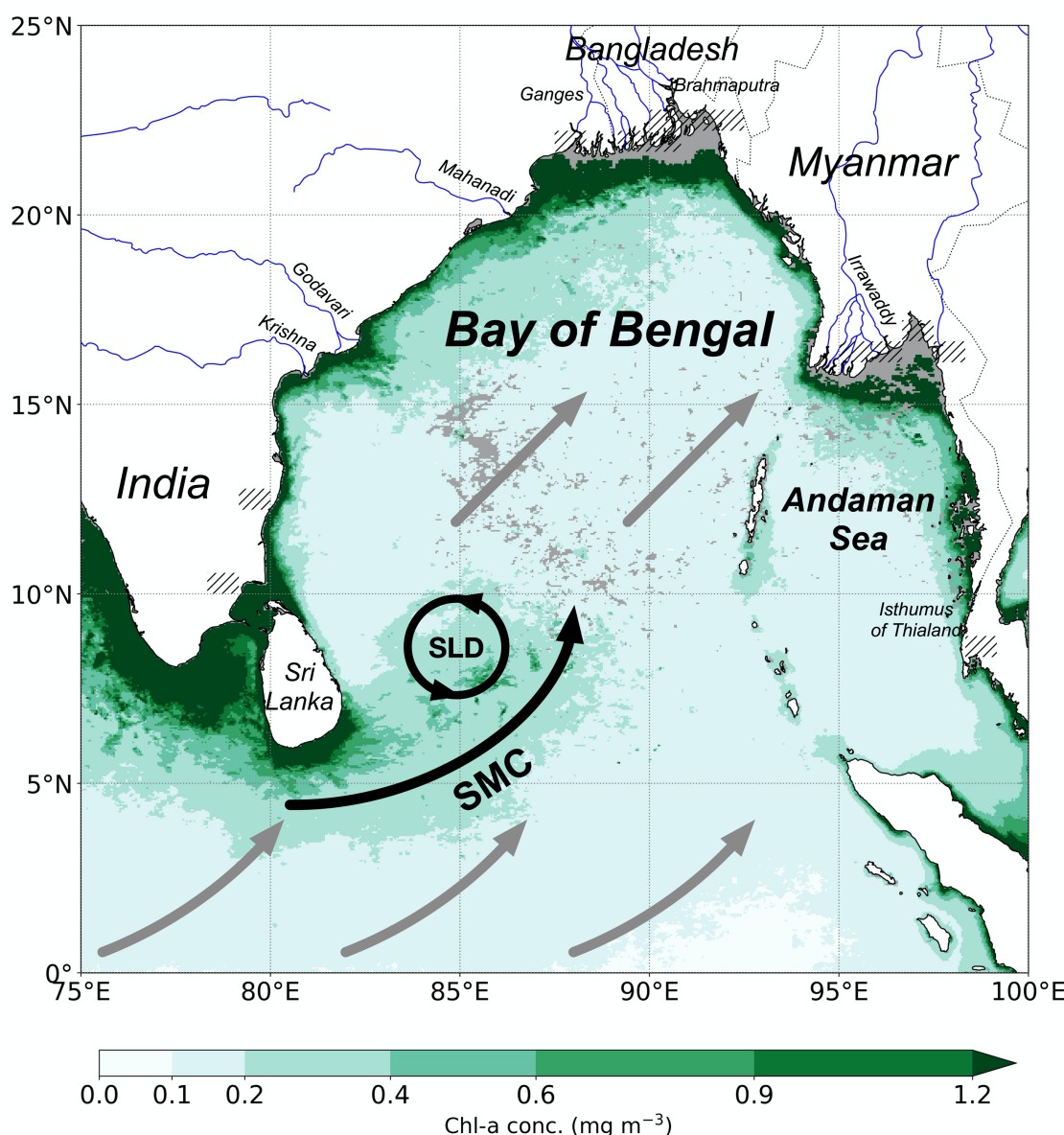

**Figure 1: The Bay of Bengal (BoB) and surrounding region of interest.** Average JJAS chlorophyll-*a* concentration climatology measured from MODIS-Aqua at 4 km horizontal resolution is shown. The locations of major rivers are represented as blue lines. The Sri Lanka Dome (SLD) is shown as a cyclonic (anticlockwise) black circle and the Southwest Monsoon Current (SMC) is shown as the solid black arrow. South westerly monsoon winds are shown as the solid grey arrows. Missing chlorophyll concentration data is shown in grey. Location of missing $h_2$ grid points in MC-KPP are shown by the black hatching.

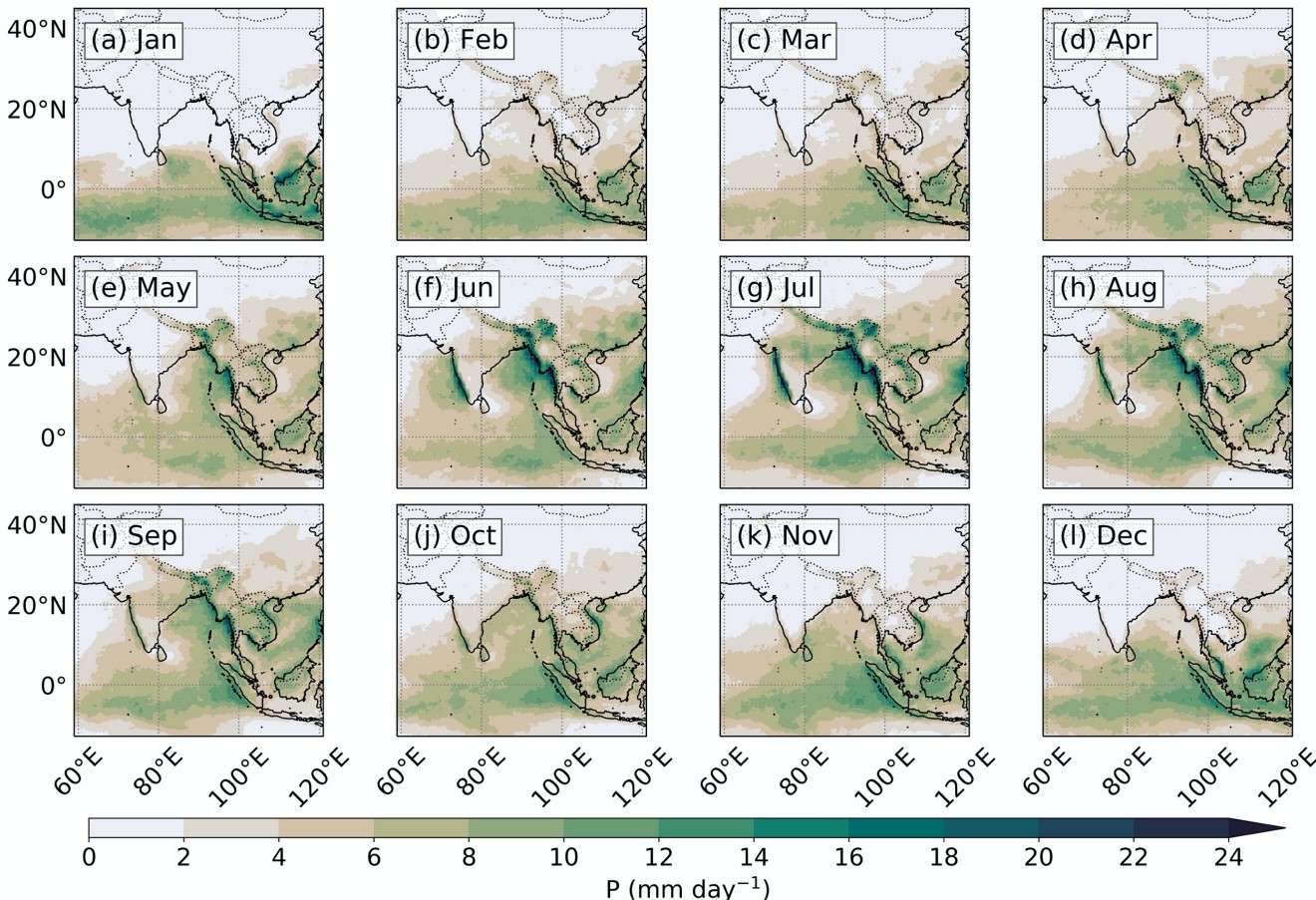

**Figure 2: Monthly climatological precipitation rate measured from the TRMM 3B42 satellite product from January to December.**


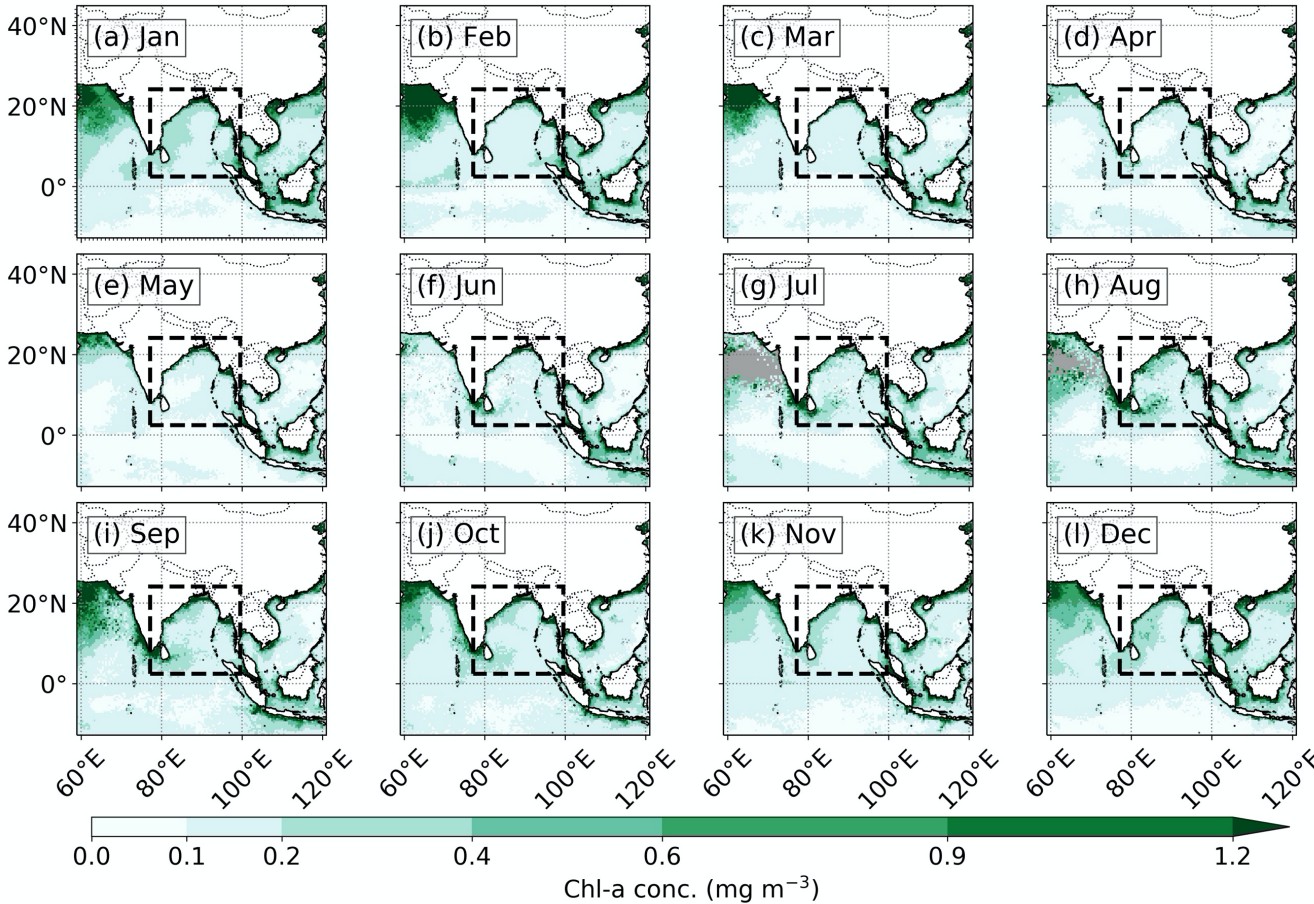

**Figure 3: Monthly chlorophyll-*a* concentration climatology measured from MODIS-Aqua at 4 km horizontal resolution from January to December. The BoB domain is outlined by a black dashed box (77–99.5° E, 2.5–24° N), which shows the location of the imposed annual cycle of chlorophyll concentration for the perturbation simulation. Missing data is shown in grey.**


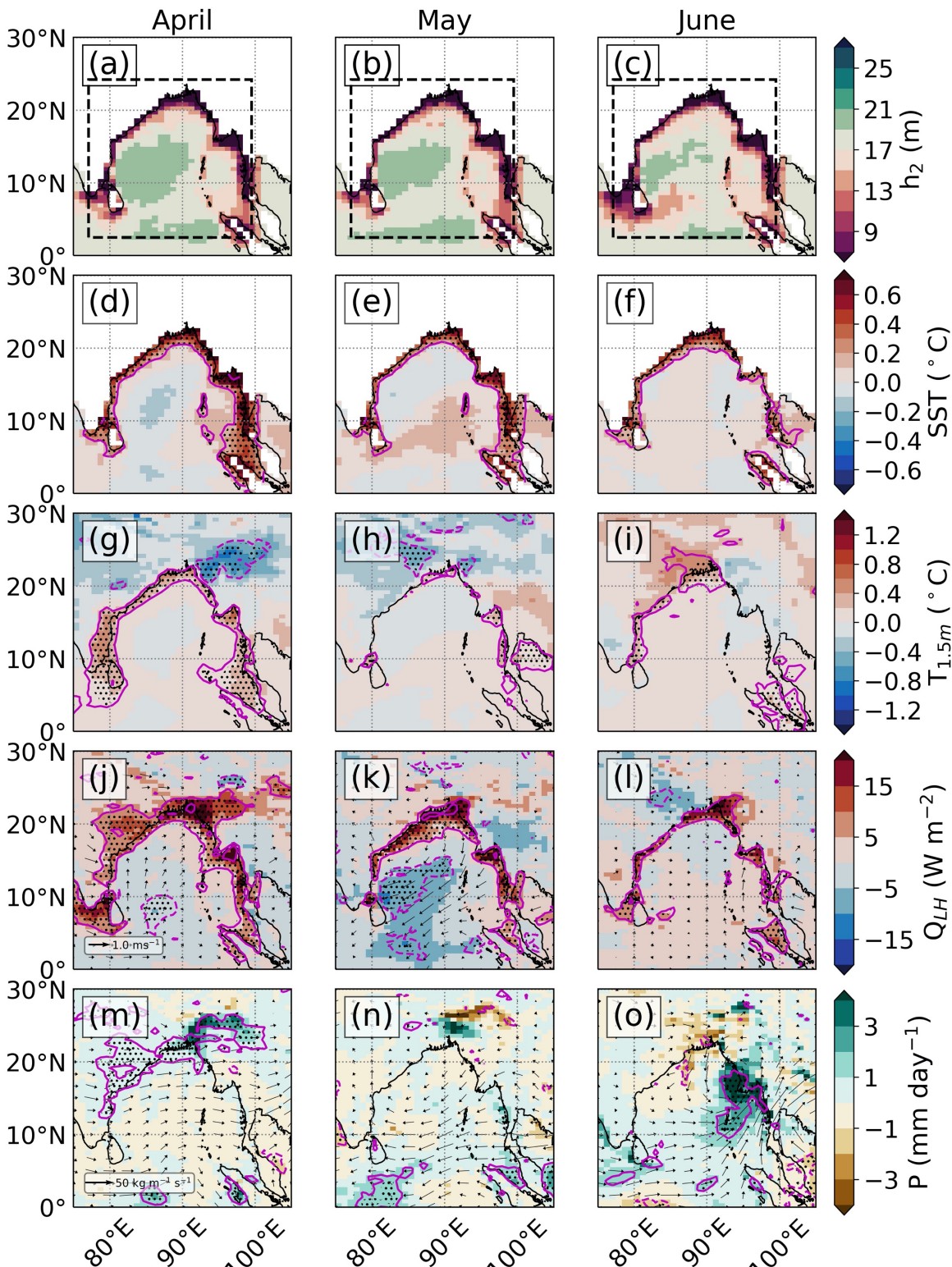

Figure 4: Monsoon onset season (April to June). (a-c) Monthly average $h_2$ (m) in the perturbation run. Monthly 29-year average difference (perturbation minus control) of: (d-f) SST (°C); (g-i) 1.5 m air temperature (°C); (j-l) upward latent heat flux (W m$^{-2}$) and 10 m wind velocity (m s$^{-1}$); (m-o) precipitation rate (mm day$^{-1}$). The magenta line shows the 10% significance level and the black stippling shows the 5% significance level.

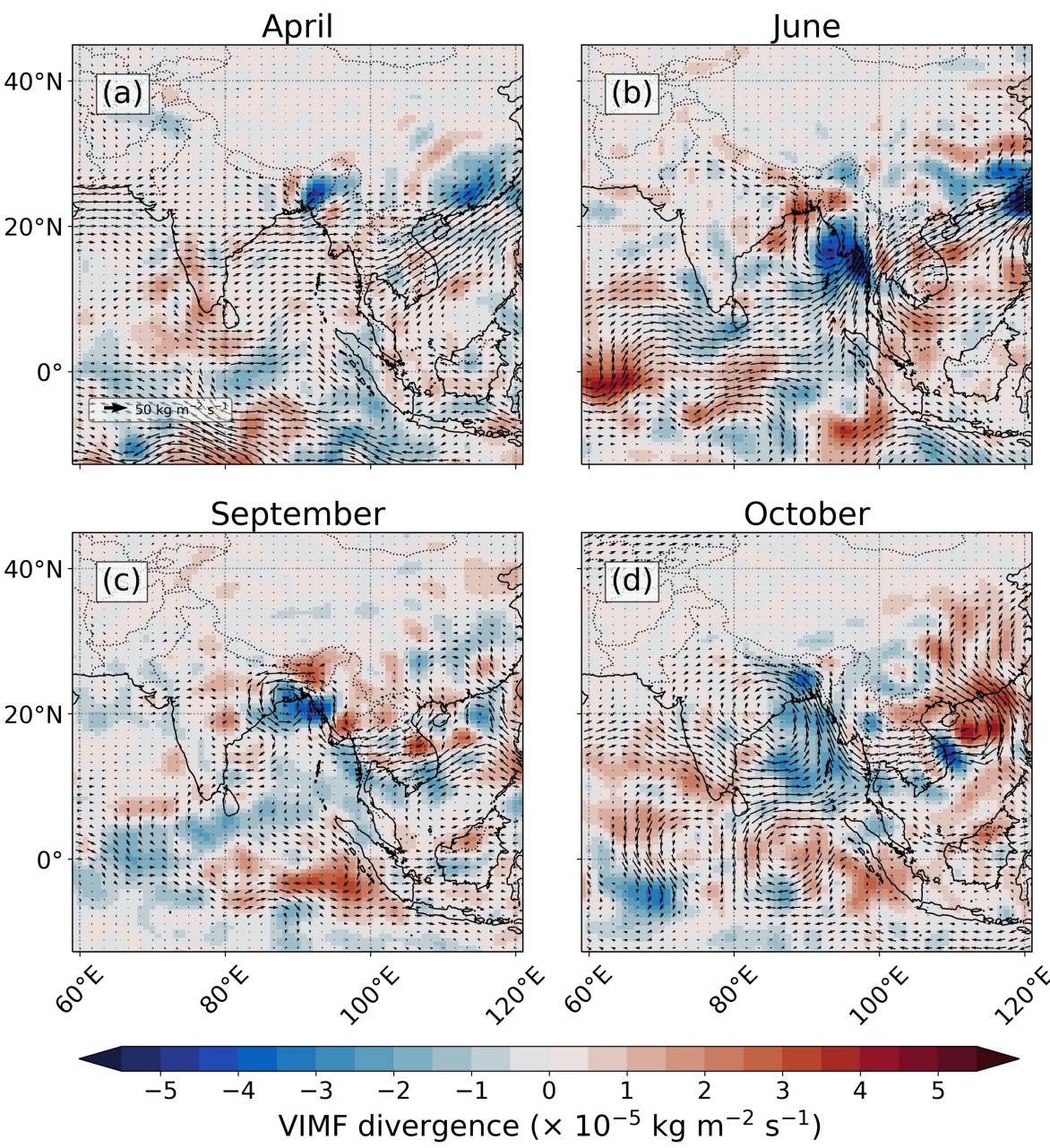

**Figure 5: Monthly 29-year average difference (perturbation minus control) of VIMF (vector arrows) and VIMF divergence (shaded) for: (a) April; (b) June; (c) September; (d) October.**



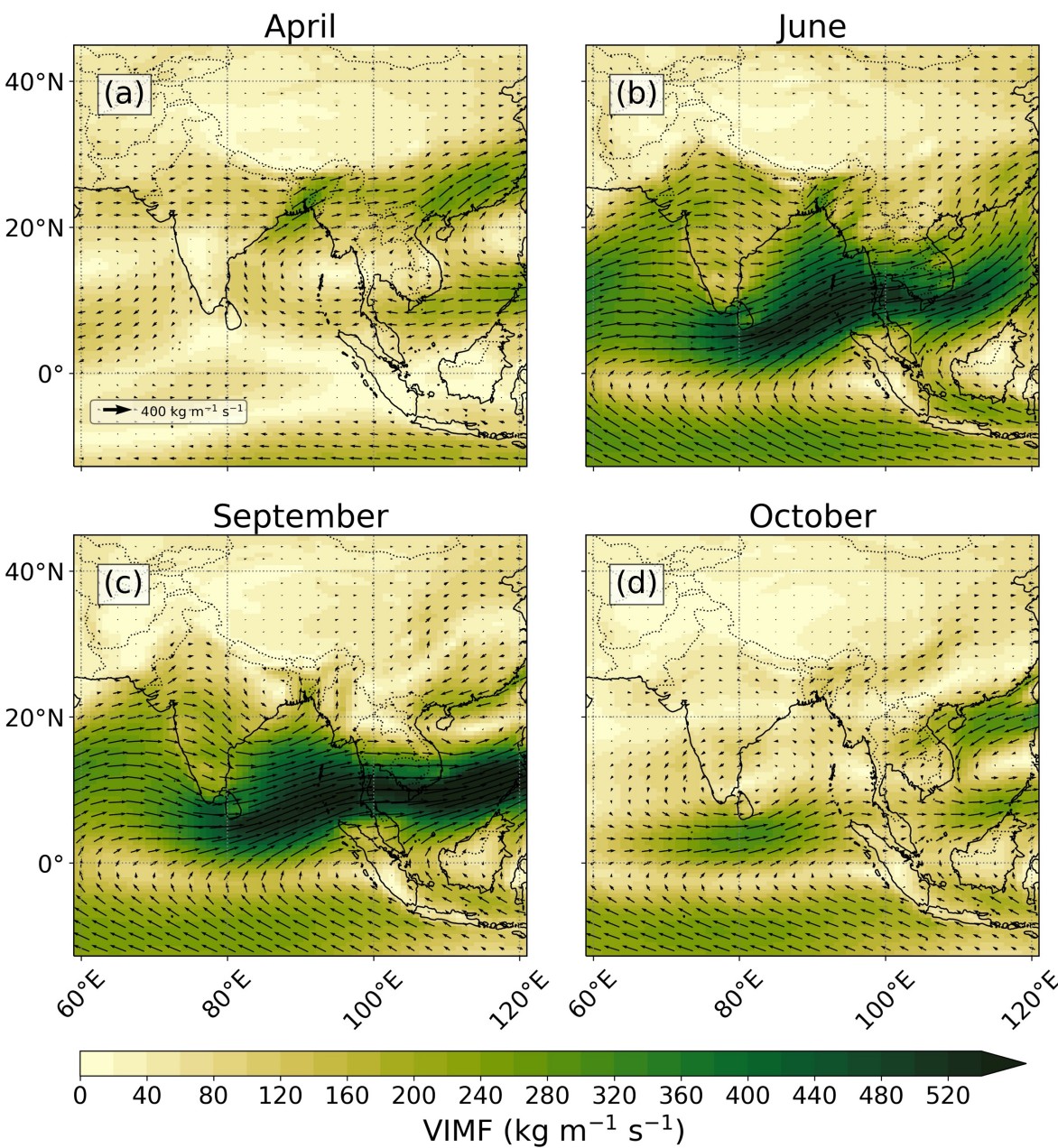

**Figure 6: Monthly 29-year average VIMF from the control run for: (a) April; (b) June; (c) September; (d) October.**

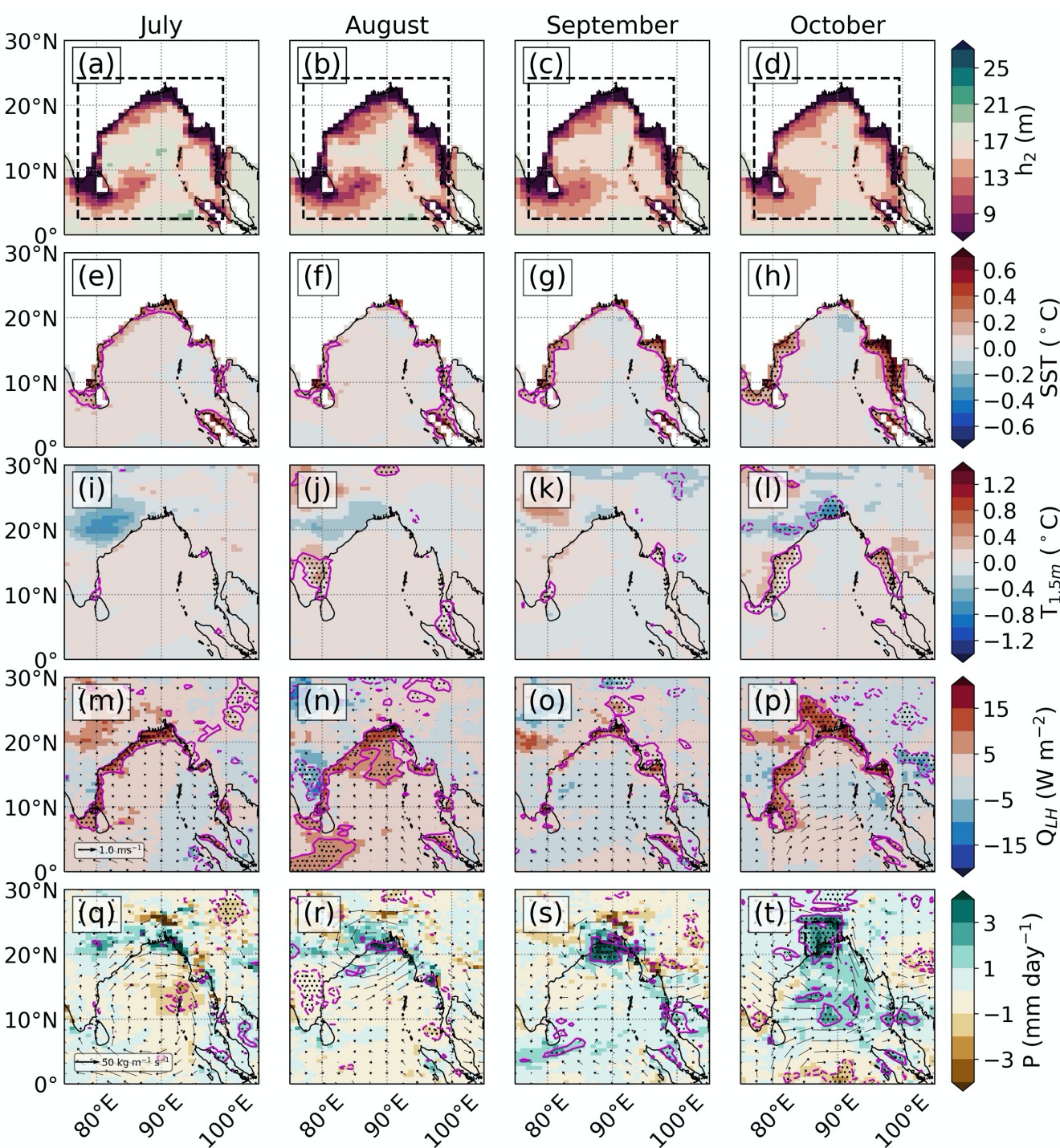

**Figure 7: As Figure 4 but for the southwest monsoon season (July to October). (t) The location of the monthly 29-year area-weighted average precipitation rate in Figure 9 is shown as a black dashed box (85–90° E, 20–25° N).**

895

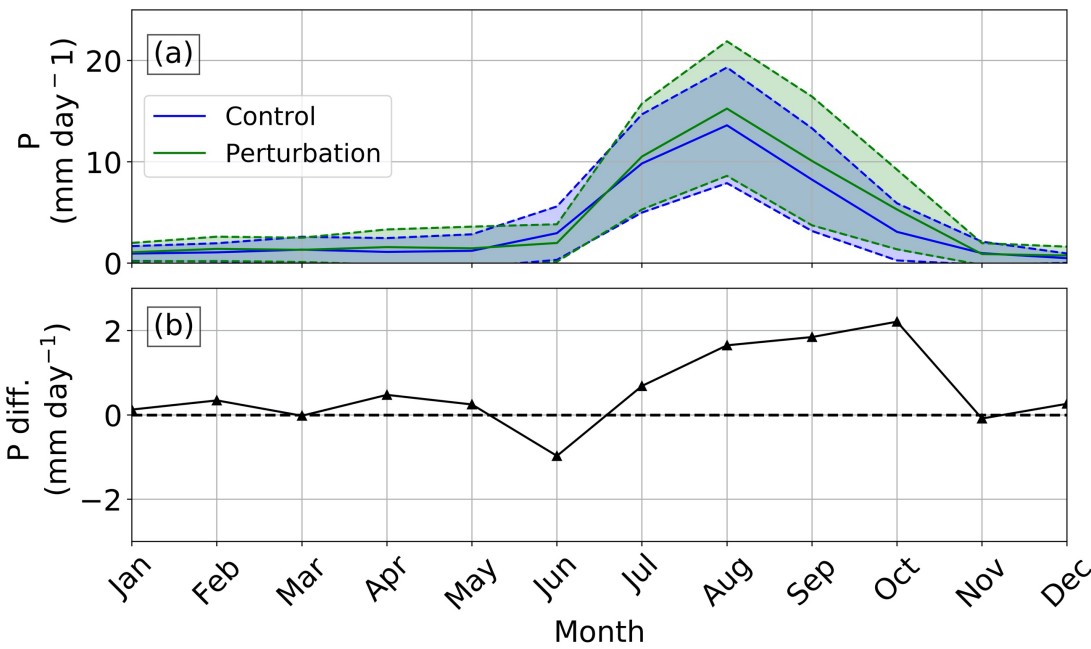

**Figure 8: (a) Monthly 29-year area-weighted average precipitation rate for the control run (blue solid line) and the perturbation run (green solid line) for the region 85–90° E, 20–25° N. Shaded region between the dashed lines shows the one standard deviation variability. (b) The difference between the monthly 29-year area-weighted average precipitation rate between the control and perturbation run.**

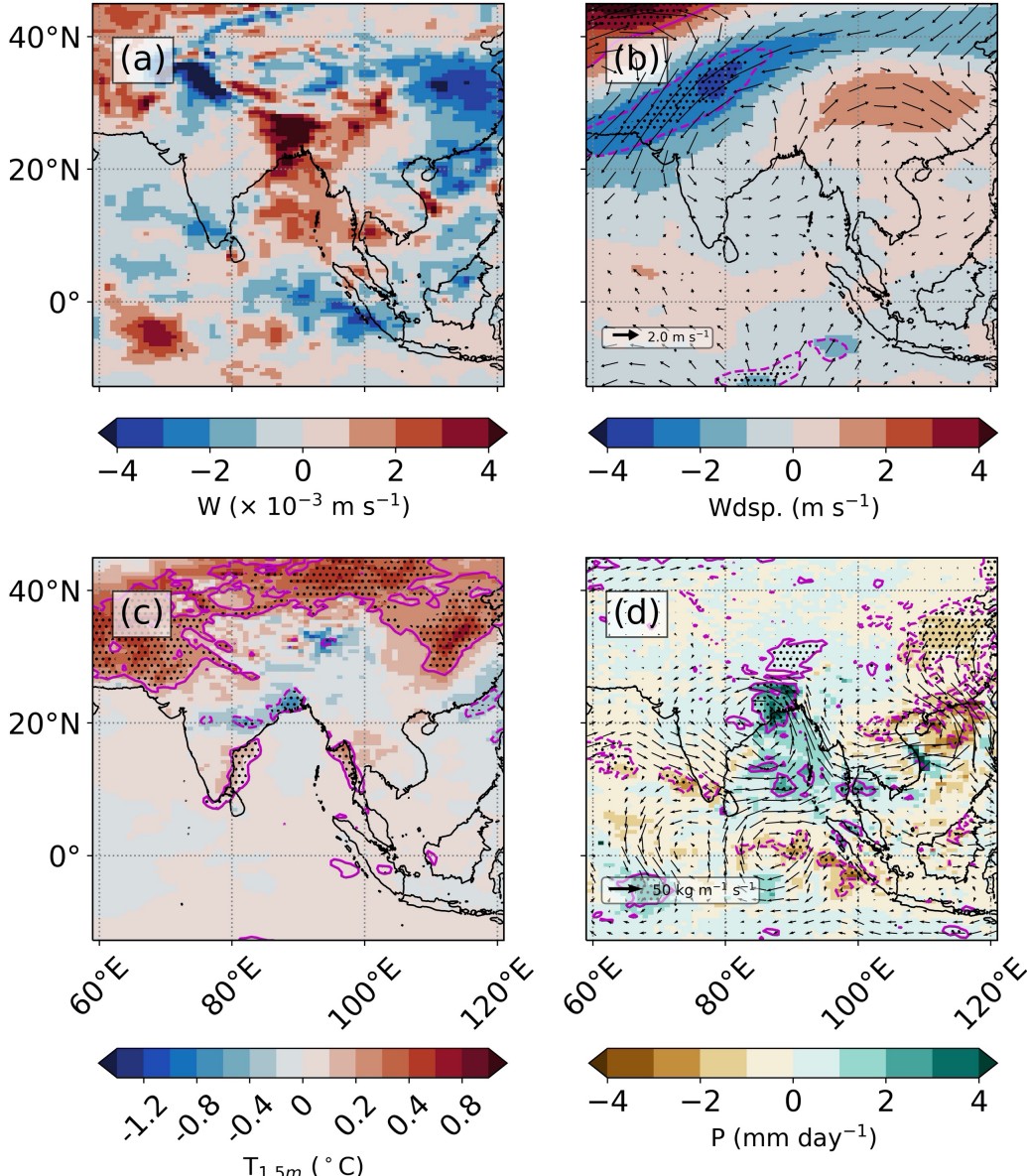

**Figure 9: October 29-year average difference (perturbation minus control) of: (a) 500 hPa vertical velocity (m s⁻¹); (b) 200 hPa horizontal vector wind (m s⁻¹); (c) 1.5 m air temperature (°C); (d) precipitation rate (mm day⁻¹) and VIMF (kg m⁻¹ s⁻¹). The magenta line shows the 10% significance level and the black stippling shows the 5% significance level.**

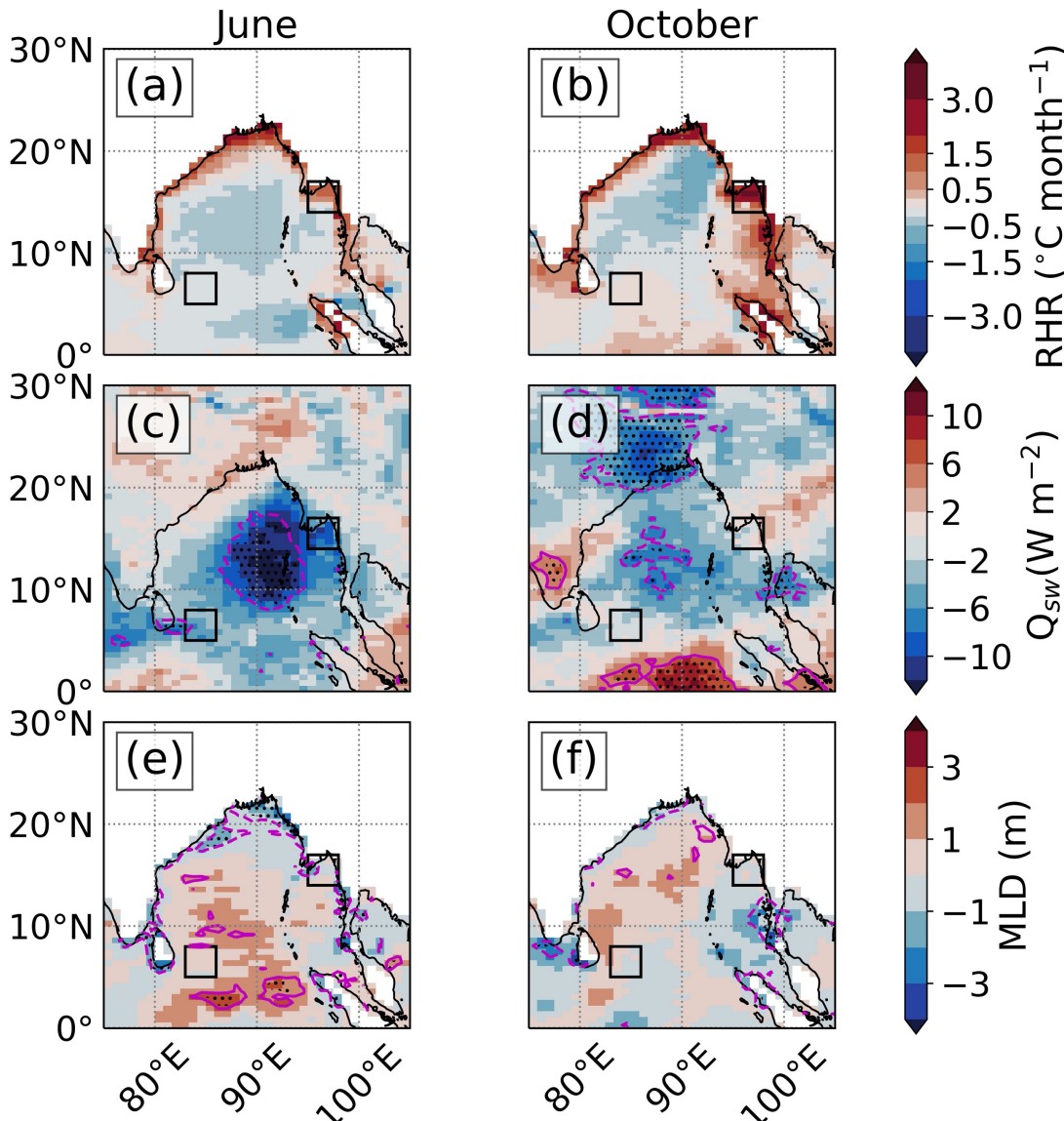

**Figure 10: Monthly 29-year average difference (perturbation minus control) for June and October of: (a,b) radiant heating rate (°C month⁻¹); (c,d) downward shortwave radiation flux (W m⁻²); (e,f) mixed layer depth (m). The black boxes show the location of the open ocean region of the SMC (southwest BoB; 83–86° E, 5–8° N) and the coastal region of the Irrawaddy Delta (northeast BoB; 95–98° E, 14–17° N). The magenta line shows the 10% significance level and the black stippling shows the 5% significance level.**





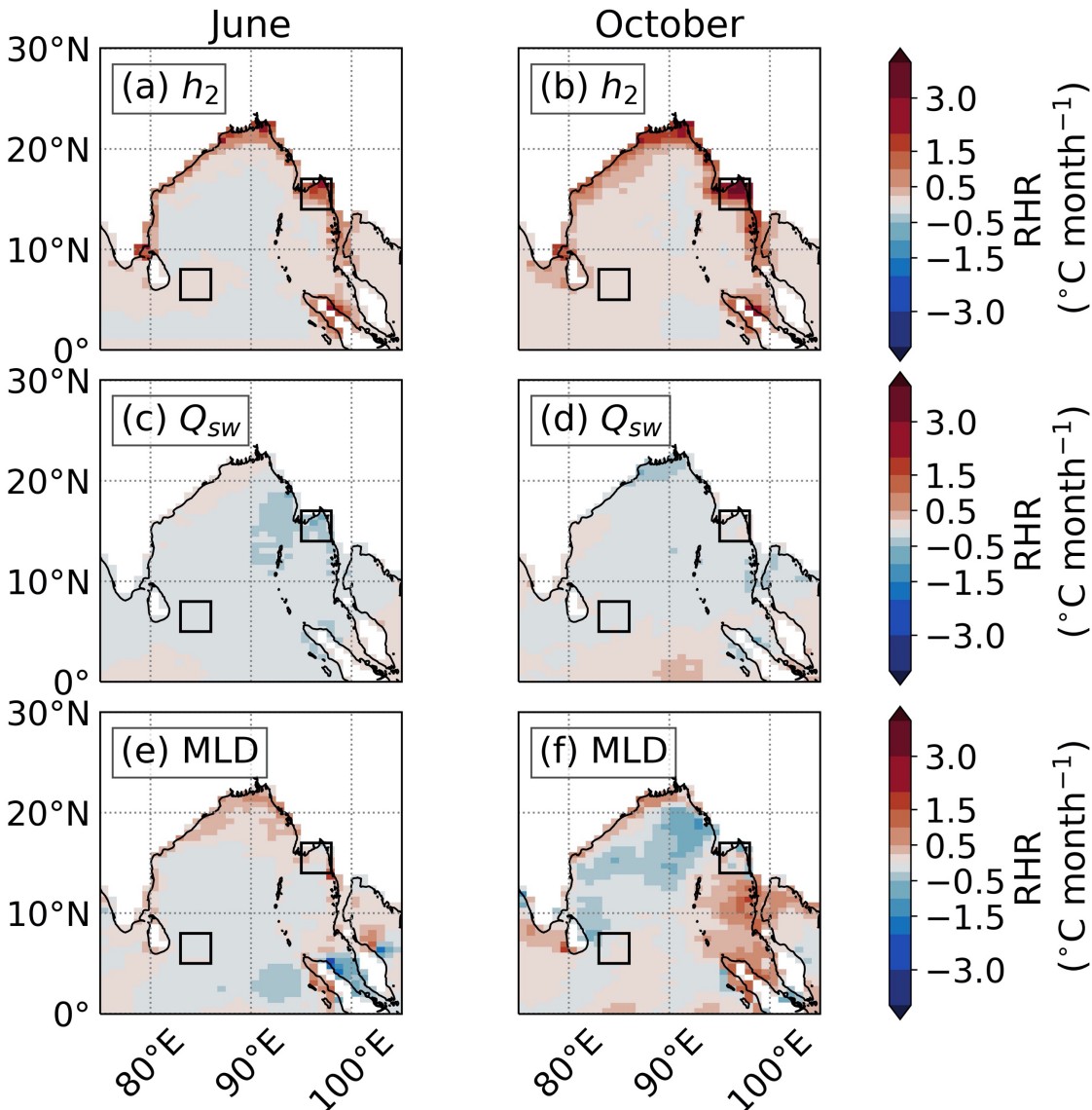

Figure 11: Monthly 29-year average difference (perturbation minus control) for June and October of the estimated relative contribution to changes in the radiant heating rate (°C month$^{-1}$) from: (a,b) $h_2$; (c,d) downward shortwave radiation flux; (e,f) MLD. As in Figure 10, the black boxes show the location of the open ocean SMC region and the coastal Irrawaddy Delta region.

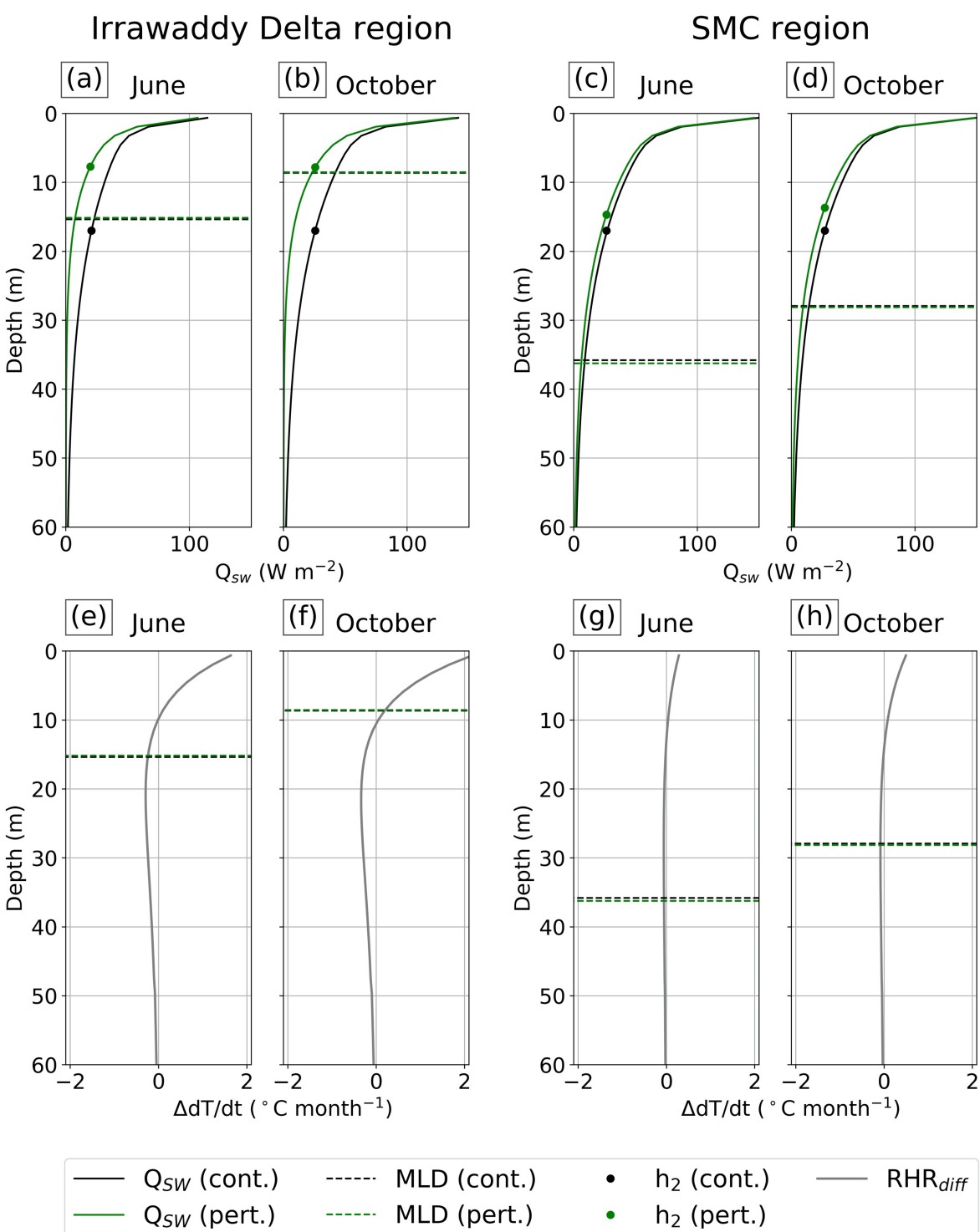

**Figure 12:** Top panels show vertical profiles of downward shortwave radiation flux from 0 to 60 m for the control (black) and perturbation (green) run for the Irrawaddy Delta region and SMC region during: (a,c) June; (b,d) October. Bottom panels show vertical profiles of radiant heating rate difference from 0 to 60 m during: (e,g) June; (f,h) October. Dashed lines show the area-weighted 29-year average mixed layer depth and coloured dots show the area-weighted average scale depth.

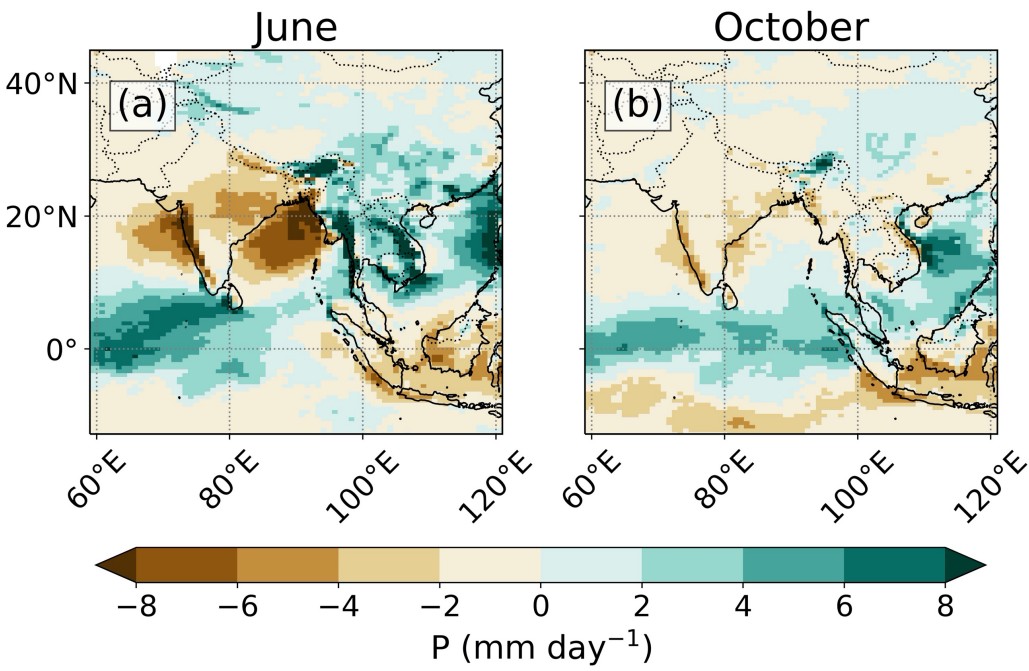

**Figure 13: Model bias of precipitation rate for: (a) June; (b) October. Bias calculated as the monthly 29-year average precipitation rate from the control run minus the monthly climatological precipitation rate observed from TRMM satellite.**

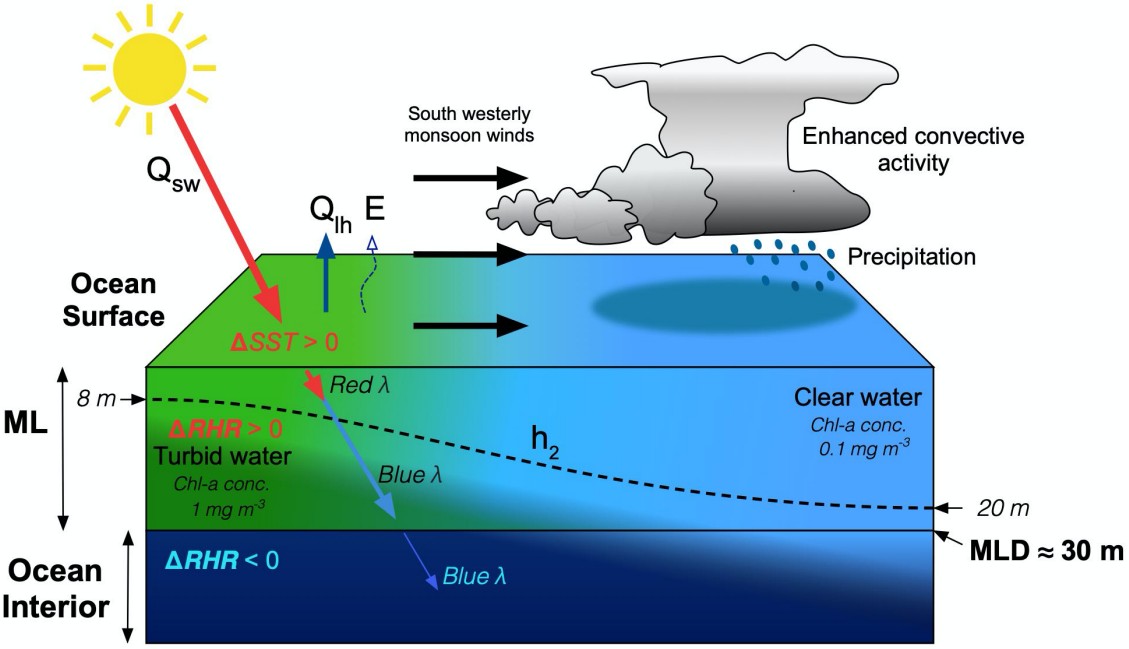

985

**Figure 14: Schematic of the effect of chlorophyll-induced heating on monsoon rainfall in the interior of the BoB.** High chlorophyll concentration in the mixed layer (ML) affects the penetration of shortwave radiative heat flux ($Q_{sw}$), the scale depth of blue light ($h_2$), and the difference in mixed layer radiant heating rate ($\Delta$RHR) and SST ($\Delta$SST) relative to clear water, which further affects

990 the surface latent heat flux ($Q_{lh}$) and evaporation ($E$). The thick red and blue arrows pointing downwards in the mixed layer illustrates the preferential absorption of the shallow penetrating red light and the deep penetrating blue light. The thin blue arrow pointing downwards below the mixed layer shows the small fraction of penetrative blue light below the mixed layer. The dashed black line in the mixed layer represents $h_2$. The three solid black arrows across the ocean surface represents the south westerly monsoon winds transporting heat and moisture that sustains enhanced convection and precipitation over the interior BoB.

995