# Peer review of "The effect of seasonally and spatially varying chlorophyll on Bay of Bengal surface ocean properties and the South Asian Monsoon"

_Weather and Climate Dynamics, 2020_

## Referee Comment (RC1) · Anonymous Referee #1 · 19 Jun 2020

General comments

Giddings et al. used an atmospheric general circulation model coupled to an ocean mixed layer model to assess the impact of spatial and seasonal variability of biological radiative heating on the seasonal monsoon in the Bay of Bengal (BoB). Two experiments were performed, one in which the attenuation depth of visible radiation is constant, the other in which the attenuation for visible radiation is spatially and seasonally varying based on a monthly climatology of satellite chlorophyll measurements. The authors find that imposing spatially- and seasonally-varying attenuation depths gives rise to modified patterns of sea surface temperature (SST) and moisture flux, which at

some location contribute to improving the precipitation biases of the model. By means of a mixed layer heat budget analysis, the authors find that the depth of visible light attenuation relative to that of the mixed layer is a crucial factor in determining whether the radiative heating will significantly affect the SST.

I find that the manuscript is promising in its methodological concept and that it could make an interesting contribution to the research field addressed by WCD. The ideas, tools and concepts are not fully novel, but their application to the South Asian monsoon has not yet been extensively elucidated in past research. However, I find that the manuscript should be improved in terms of clarity, and that the results are not conveyed in a satisfying manner.

Specific comments

In order to support the conclusions, I believe that five aspects should be improved:

1) The authors should state more clearly the aims and the key results of the study as well as their contribution to the research field. It is not clear to me whether the focus is on improving model biases or on the mechanistic understanding of how biological radiative heating affects the monsoon in the BoB.

2) In the Introduction, the physical mechanisms driving the mean seasonality of the South Asian monsoon should be better explained. Fig. 1 could be extended to contain two panels (e.g. for summer and autumn) showing schematically the prevalent winds and ocean currents associated with the monsoon. The areas of the chlorophyll peaks could be highlighted with e.g. a box.

3) I believe the figures could be adapted to make the relevant results stand out more clearly. In particular, I find Fig. 4 and Fig. 8 particularly trying. The plots are extremely small (zooming on the BoB region would certainly help) and the areas, where changes are statistically significant, are difficult to see (also because of the dots used for the political boundaries, are these needed?). In addition, the large number of figures makes

it difficult to extract the relevant information. I suggest that the authors think carefully whether all the figure panels are really necessary and whether there could be a more effective way of showing the information. An idea could be to show seasonal cycles of these variables spatially-averaged over dynamically-consistent regions (e.g. Irrawaddy Delta region/SMC region, or open ocean/coastal, or north/south) and to show the difference maps only for selected months, seasons or for a subset of variables. Having the whole seasonal cycle on one plot (as in Fig. 9) would make it easier to relate the response of the physical variables to the modified attenuation depth.

4) In describing the results, the authors should make an effort to sift through the plentiful information provided by the model simulations in order to highlight the relevant messages. Concluding each of the Results' subsections with a few summarizing sentences would help, as well as including a final schematic showing the critical mechanisms emerging from this study.

5) If I understand correctly, a key take home message of the paper is that the interplay between the biologically-driven heating and the depth of the mixed layer, as well as the timing of the biological heating with respect to the seasonality of the monsoon, determine whether phytoplankton will exert a strong or weak influence on the monsoon. In this respect, different regions behave differently because of their underlying stratification and phytoplankton seasonality. This is an intriguing point which I think the authors should expand and discuss more.

From a methodological point of view, I find it somewhat disturbing that the authors define their model as a "coupled ocean-atmosphere general circulation model (GCM)" since the ocean model is not a GCM but an ocean mixed layer configuration containing no advective processes. The opportunities and disadvantages of this methodology should be clearly expressed.

Line-by-line comments:

Lines 15-24: please state the aims of the study more clearly. Also, a few more words

could be said on the length of the simulations. As mentioned earlier, defining the model as a "coupled ocean-atmosphere model" is misleading, since it brings to the mind a coupled ocean-atmosphere GCM. I suggest being more specific.

Introduction (lines 28-104): I think the Introduction needs more structure. The fact that the paper deals with the effects of biological radiative heating on the monsoon comes quite late (line 84). My suggestion is to have a first overarching paragraph on the relevance of the monsoon and on overlooked feedbacks between biology and the monsoon. The paragraphs afterwards can go into more detail on the physical and biological properties of the region (and here, please enrich your text by referring to an updated Fig. 1 – see my point 2 of the specific comments). I also think that the contribution you want to make to the research field should be better highlighted in the last paragraph.

Lines 101-102: I believe Wetzel et al. (2006) used a coupled GCM, whereas your study actually does not. This statement is therefore incorrect.

Lines 127-128: I suggest adding a more expanded definition of h1 and h2, for instance: the e-folding depth, or attenuation depth, is the depth where surface radiation decreases by 1/e (or 37%) of its initial surface value.

Lines 138-139: Please state already here which Jerlov water type is used in the simulation with constant attenuation depth.

Lines 141-145: How does the relaxation of ocean temperature to observations affect your results? Which is the relaxation time scale and how sensitive are your results to the relaxation time scale?

Lines 155 and later line 181: why say "h2 values (i.e. chlorophyll concentration)"? It is clearly not the same quantity.

Line 173: Fig. 1 could show in gray shading the grid points in which chl-a was not determined by satellite.

[Figure]

Lines 184-185: 30-years of simulation sounds quite short, also considering that computational costs are not very high (line 149). Is the model in an adjusted state?

Lines 200-212: I think these two paragraphs should go in Section 2.2. Indeed, at the first sentence of Section 2.2 I was already wondering how you compute h2 based on chlorophyll data.

Lines 213-226: This section doesn't read well, as it contains three very different sets of information (statistical significance computation, VIMF computation, regridding of observed precipitation). Please restructure. Part of the information could maybe go in the figure captions.

Line 245: Unless I'm mistaken, Fig. 6 is cited before Fig. 5.

Lines 251-252: please explain the mechanism better: despite the SST increase, evaporation decreases because air humidity increases?

Line 255 and 265: "caused by" instead of "corresponds"?

Lines 255-259 (same for 289-291): I suggest not mixing mechanism understanding and bias improvement in one paragraph. Indeed, maybe all the part on bias improvement could be shifted to the discussion?

Lines 263-272: This paragraph is somewhat cumbersome. I suggest condensing the relevant information in fewer sentences.

Line 274: co-located?

Line 274: no comma after whereas?

Line 275: Increased SST also speeds up evaporation.

Lines 300-306: This part seems more suitable for the Discussion section.

Section 3.3 (lines 309-371): if I understand correctly, Eq. 4 is used to estimate the relative contribution of attenuation depth, incoming shortwave radiation and MLD changes

on the radiative heating. Throughout the section, you then refer to these results only in the text. I think it would be more useful to see the relative contribution of these processes in a separate figure. For instance, in Fig. 11 you could show, instead of the Qsw and MLD changes, the relative contribution that Qsw and MLD have on the RHR.

Line 320: I'm acquainted with cp=3850 J/(kg*K). Please clarify.

Line 324: "in" too much

Lines 329-330: please be more explicit on why you selected these two regions.

Lines 356-370: please state more explicitly why a deeper MLD leads to smaller biologically-driven changes in SST. If I understand correctly, the mechanism you are referring to is: an increased radiative heating in the upper levels caused by biology leads to radiative cooling in the layers below. A MLD deeper than the attenuation depth causes this dipole to mix, and therefore (assuming constant Qsw) the net effect of biological heating on ML temperature should be null. To that one should also add the fact that higher surface SSTs lead to increased ocean heat losses by evaporation. Therefore, on annual average the areas of high biology (and deep MLD) should actually experience surface cooling. I wonder whether you see this effect in your model?

Lines 356-371: It is intriguing to see that the presence of biology and associated radiative heating do not modify the MLD. Could you comment on that?

Discussion-Conclusions: The authors could consider joining the Discussion and the Conclusions. My suggestion: first start with a paragraph summarizing the main results (as in lines 435-455), then continue with the discussion points, and finally conclude with the open questions and outlook. As it is now, some parts of the Conclusions seem more apt to be in the Discussion.

Lines 388-389: The MLD changes of 1 m cannot readily be seen in Fig. 11. It would be more useful to show spatially averaged MLD anomalies as suggested in point 3 of my specific comments.

Lines 390-391: This is an interesting statement, which would imply that in coastal regions the local effect of high chlorophyll dominates the perturbation minus control anomalies in stratification, whereas in the open ocean remote effects through wind speed changes play a leading role. Some more analysis should be shown to substantiate this statement.

Lines 423-425: The discussion on the effect of subseasonal chlorophyll variability on the Boreal Summer Intraseasonal Oscillation comes somewhat out of the blue, since you have not mentioned any form of intraseasonal or interannual variability in the paper. Please introduce this paragraph better to put the reader in the picture.

Lines 430-431: Which feedbacks and processes are you missing by using an ocean mixed layer model instead of an ocean GCM (and by using an imposed chlorophyll distribution instead of an interactive biogeochemical model)? Do you expect the results from your modeling setting to be rather overestimating or underestimating the "true" response?

Line 436: Please be more specific on the model you used. (Coupled ocean-atmosphere GCM is misleading).

Lines 450-454: The inclusion of biological heating reduces the precipitation biases in some locations, which is good, but does it increase the biases elsewhere? (e.g. over Myanmar?)
* * *

---

## Short Comment (SC1) · 6 Jul 2020

Short response to Reviewer 1

Thank you for your constructive comments and suggested improvements of our manuscript. We believe your comments will help to improve our manuscript. We feel we are able to address all the specific and line-by-line comments. In this short response, we will provide a brief reply to the five specific comments made by Reviewer 1, which may help other reviewers with similar concerns. A full response will be submitted after receiving all reviewers comments.

[Figure]

The first specific comment by Reviewer 1 was the need for clarity of the aims of the study. The aim of the present study is to understand how biological radiative heating affects the monsoon in the BoB. The model bias is used to further understand how the chlorophyll-induced signal affects monsoon rainfall over the BoB. In the revised manuscript, we will make sure that the aim, purpose and contribution of the study are stated more clearly throughout.

The second specific comment was the need for further explanation of the physical mechanisms of the mean seasonality of the South Asian monsoon and to show these mechanisms schematically on Fig.1. In the revised manuscript, we will make sure to add a short section explaining the physical mechanisms of the South Asian monsoon in the Introduction section. We will further include relevant oceanic currents and atmospheric winds schematically on Fig. 1 to better convey the key physical mechanisms of the South Asian monsoon.

The third specific comment referred to Fig. 4 and Fig. 8 in the Results section. We acknowledge that these figures are particularly small. We agree that the latitudinal and longitudinal domain size should be reduced, and the political boundaries should be removed to significantly improve the extraction of relevant information from Fig. 4 and Fig. 8. As we wait for other reviewers comments, we will think carefully about how to display the relevant information more clearly in the manuscript.

The third specific comment was the need to include summarising sentences at the end of the Results' subsections. We will make sure summarising sentences are added to conclude the main findings of each Results subsection. Fig. 1 submitted with this short response (see page C4) shows a schematic that we hope summarises the main findings of the study and will be added to the revised manuscript.

The final specific comment by Reviewer 1 was the need to expand and discuss the take home message of the paper. Reviewer 1 is correct about the take home message and we will aim to convey this message more clearly in the Discussion/Conclusions

sections of the revised manuscript.

We again appreciate the specific and line-by-line comments made by Reviewer 1 and look forward to receiving other reviews in the near future.
* * *
[Figure]

[Figure]

**Fig. 1.** Schematic of the penetration of shortwave radiative heat flux (Qsw), scale depth of blue light (h2), mixed layer radiant heating rates (dT/dt), change in SST (△SST) and convective rainfall.

---

## Referee Comment (RC2) · Anonymous Referee #2 · 11 Jul 2020

This paper is well written, well motivated and timely. Understanding the impact of surface chlorophyll on the SST is a new emerging area of satellite oceanography and a multidisciplinary dynamical issue that needs attention. Application of coupled ocean-atmosphere model is thus a reasonable approach using observed chlorophyll data to force the coupled system and understand the sensitivity of SST response to changes in the chlorophyll behavior.

It would be useful if the authors would comment on the following aspects of their simulation and interpretation.

1. It would be good to quantify the impact of chlorophyll on the changes in SST. Maybe

the Abstract or the Conclusion could capture this important aspect.

2. Could the authors clarify the errors in their estimates of impact, given that the horizontal resolution of the effective coupled model is actually governed by the 90 km grid spacing of the MetUM-GOML3.0? I understand that the 4km satellite observations have been re-gridded to the 90km model grid. If this is not correct, could the authors specify the horizontal resolution of the ocean model (which is really okay for the vertical – with 100 levels in the upper 1000 m, of which 70 levels are in the top 300m)?

3. Given the 90 km horizontal resolution of the ocean model, how reliable are the inferences for the "coastal chlorophyll impacting the SST" results? Is there really a coast in the ocean model? Should we interpret them as near-coastal? The whole BOB would be a 25 x 25 grid ocean at 90 km resolution. Most of the coastal regions are resolved by such a grid with less than 2 grid points. Maybe the authors could clarify this with care, so that future studies can build upon this limitation.

4. Furthermore, as authors point out in lines 205-212, there are lots of missing values of h2 in coastal regions (not surprisingly) – this again could influence the inferences in lines 384-400. Please also see comment 7 below.

5. The first paragraphs of Section 3.1 and 4.1 are hard to follow. This reader was getting very confused with the increase/decrease and high/low ordering of sentences. Maybe talk about changes focusing on regions first and then the months. A minor issue is the frequent usage of the phrase 'in reality' in both paragraphs. This phrase occurs five times in lines (233-239) and another five times in lines (264-271) – almost once in every sentence. Maybe use 'observed', 'satellite' etc.

6. Lines 245, 257 – the superscripts for units did not come through in my downloaded version. There are other similar occurrences throughout the paper. Please check.

7. It is encouraging to note that the authors have used the Satellite-derived chlorophyll concentrations to h2 using a fifth-order polynomial parameterization. They reported the
improvement of SST and precipitation in the coastal region. However, the coastal BoB is mostly dominated by river water, where the above algorithm (5th order polynomial to get h2) might not be totally applicable. The authors have actually interpolated and/or extrapolated to fill the data gaps (page 5, 205). This could possibly lead to a positive bias in precipitation in the head BoB in the post-monsoon period (Figure 8). Could the authors please clarify these two aspects or limitations.

8. The BoB is a highly eddy-active region, which has a significant impact on the chlorophyll distribution and on the air-sea interaction (page 13, 570). The authors could expand on how finer resolution ocean models might be helpful in the future for resolving both eddy activities in the open region and mesoscale to sub-mesoscale features in coastal BOB.

9. In the BoB (like in other oceans), Chlorophyll maxima is generally not at the surface level. It varies from 10 m to 80 m (Pramanik et al., 2020). The impact of the deep chlorophyll maxima and its relationship with the surface chrolophyll and SST variations may be explored in a separate future work, but worth a mention.

Reference:

Pramanik, S., Sil, S., Gangopadhyay, A., Singh, M.K. and Behera, N., 2020. Interannual variability of the Chlorophyll-a concentration over Sri Lankan Dome in the Bay of Bengal. International Journal of Remote Sensing, pp.1-18.

---

## Author Comment (AC1) · 15 Sep 2020

**Author Comments 1 on "The effect of seasonally and spatially varying chlorophyll on Bay of Bengal surface ocean properties and the South Asian Monsoon"**

We would like to thank Reviewer 1 who provided constructive and insightful comments that have greatly improved the revised manuscript. We have incorporated all their suggestions where possible. Reviewer 1 comments have been reproduced in black with authors response in blue and excerpts from the revised manuscript in italics. The revised and renumbered figures are included at the end of the document.

**Response to Reviewer 1**

**Specific Comments**

1. The authors should state more clearly the aims and the key results of the study as well as their contribution to the research field. It is not clear to me whether the focus is on improving model biases or on the mechanistic understanding of how biological radiative heating affects the monsoon in the BoB.

The aim of the present study is to understand how biological radiative heating affects the monsoon in the BoB. The model biases are reduced with a more accurate representation of chlorophyll, highlighting the importance of including chlorophyll in coupled models. The motivation and aim of the study are now stated more clearly in the Introduction.

"Although the effect of chlorophyll on BoB SST has been demonstrated by these previous studies, the effect of chlorophyll on monsoon rainfall remains a vital knowledge gap. Without this knowledge, missing bio-physical interactions in the BoB could lead to inaccuracies in simulated air-sea interactions that are crucial in representing accurate monsoon behaviour and thus rainfall timing, location and duration over the Indian subcontinent."

The contribution of the study is now also stated more clearly in the Discussion and Conclusions section.

2. In the Introduction, the physical mechanisms driving the mean seasonality of the South Asian monsoon should be better explained. Fig. 1 could be extended to contain two panels (e.g. for summer and autumn) showing schematically the prevalent winds and ocean currents associated with the monsoon. The areas of the chlorophyll peaks could be highlighted with e.g. a box. We have added a short section explaining the basic physical mechanisms of the South Asian monsoon in the Introduction.

"The South Asian monsoon is initiated when lower-tropospheric winds, transporting heat and moisture, begin to flow northward from the equator to the Asian continent in response to increasing summer insolation and increasing land-sea thermal and pressure gradients (Grey arrows; Fig. 1; Webster et al., 1998). Mid-tropospheric heating from the elevated Tibetan Plateau increases the land-sea thermal and pressure contrast, further regulating the seasonal reversal of the large-scale circulation (Li and Yanai, 1996)." We have now included relevant oceanic currents and atmospheric winds schematically on Fig. 1 (included at the end of this document) to better convey the key physical mechanisms of the South Asian monsoon during JJAS. Fig. 1 now includes the average JJAS chlorophyll concentration from MODIS-Aqua. We did not include a separate panel for Autumn as the atmospheric circulation features are captured in later figures. The suggestion to include boxes to highlight key chlorophyll features is a good one which we experimented with. However, we did not include these in the end as they were found to clutter the figure, which is now quite busy with wind and ocean current information added.

3. I believe the figures could be adapted to make the relevant results stand out more clearly. In particular, I find Fig. 4 and Fig. 8 particularly trying. The plots are extremely small (zooming on the BoB region would certainly help) and the areas, where changes are statistically significant, are difficult to see (also because of the dots used for the political boundaries, are these needed?). In addition, the large number of figures makes it difficult to extract the relevant information. I suggest that the authors think carefully whether all the figure panels are really necessary and whether there could be a more effective way of showing the information. An idea could be to show seasonal cycles of these variables spatially-averaged over dynamically-consistent regions (e.g. Irrawaddy Delta region/SMC region, or open ocean/coastal, or north/south) and to show the difference maps only for selected months, seasons or for a subset of variables. Having the whole seasonal cycle on one plot (as in Fig. 9) would make it easier to relate the response of the physical variables to the modified attenuation depth.

We thank the reviewer for these helpful suggestions, most of which we have adopted to improve Figs. 4, 7, 10 and 11 (Please note Fig. 8 has now been renumbered to Fig. 7). The domain has been reduced to  $75-105^{\circ}$ E,  $0-30^{\circ}$ N and the political boundaries have been removed to significantly improve the extraction of relevant information. The revised and renumbered Figures are shown at the end of this document.

4. In describing the results, the authors should make an effort to sift through the plentiful information provided by the model simulations in order to highlight the relevant messages. Concluding each of the Results' subsections with a few summarizing sentences would help, as well as including a final schematic showing the critical mechanisms emerging from this study. This is a good suggestion. We have now included summarising sentences to conclude the main findings of each Results subsection.

"We have thus demonstrated that high coastal chlorophyll concentrations in spring perturb the absorption of solar radiation, which increases air temperature and SST, and in turn significantly increases spring intermonsoon precipitation rates in the northern and eastern BoB."

"As in the spring intermonsoon, the increase in precipitation rate during autumn intermonsoon in the northern BoB is primarily attributed to high coastal chlorophyll concentrations and associated increased SST extending from the Andaman Sea to the Ganges river delta along the chlorophyll-perturbed BoB coastal region."

Fig. 14 at the end of this document shows a schematic that we hope summarises the main findings of the study. The Figure is mentioned in the Discussion and Conclusions section.

5. If I understand correctly, a key take home message of the paper is that the interplay between the biologically-driven heating and the depth of the mixed layer, as well as the timing of the biological heating with respect to the seasonality of the monsoon, determine whether phytoplankton will exert a strong or weak influence on the monsoon. In this respect, different regions behave differently because of their underlying stratification and phytoplankton seasonality. This is an intriguing point which I think the authors should expand and discuss more.

Reviewer 1 is correct about the take home message. We have now endeavoured to convey this message more clearly and have expanded the discussion of this point in the Discussion and Conclusions section.

6. From a methodological point of view, I find it somewhat disturbing that the authors define their model as a "coupled ocean-atmosphere general circulation model (GCM)" since the ocean model is not a GCM but an ocean mixed layer configuration containing no advective processes. The opportunities and disadvantages of this methodology should be clearly expressed.

We believe that it is appropriate to refer to MetUM-GOML as a GCM, in line with previous studies using this configuration (e.g., Hirons et al., 2015; Peatman and Klingaman, 2018). MetUM-GOML includes a state-of-the-art atmospheric GCM, coupled to a simplified ocean mixed-layer model. The reviewer is correct that the MetUM-GOML ocean does not simulate ocean dynamics directly, but the mean effect of dynamics on the ocean is parameterised through the temperature and salinity corrections applied. In deference to the reviewer's request, we have replaced "GCM" with "MetUM-GOML". The limitations of using MetUM-GOML are discussed on Lines 430-431 and specific comments 2, 3 and 8 from Reviewer 2.

**Line-by-line comments:**

Lines 15-24: please state the aims of the study more clearly. Also, a few more words could be said on the length of the simulations. As mentioned earlier, defining the model as a "coupled ocean-atmosphere model" is misleading, since it brings to the mind a coupled ocean-atmosphere GCM. I suggest being more specific.

The motivation of the study has been stated more clearly in the Abstract as suggested.

"Although the influence of chlorophyll distributions in the Arabian Sea on the southwest monsoon has been demonstrated, there is a current knowledge gap in how chlorophyll distributions in the Bay of Bengal influence the southwest monsoon."

The study aim, model type and length of simulations have been specified in the Abstract.

"Seasonally and spatially varying  $h_2$  fields in the Bay of Bengal were imposed in a 30year simulation using an atmospheric general circulation model coupled to a mixed layer thermodynamic ocean model to investigate the effect of chlorophyll distributions on regional SST, southwest monsoon circulation and precipitation."

Introduction (lines 28-104): I think the Introduction needs more structure. The fact that the paper deals with the effects of biological radiative heating on the monsoon comes quite late (line 84). My suggestion is to have a first overarching paragraph on the relevance of the monsoon and on overlooked feedbacks between biology and the monsoon. The paragraphs afterwards can go into more detail on the physical and biological properties of the region (and here, please enrich your text by referring to an updated Fig. 1 - see my point 2 of the specific comments). I also think that the contribution you want to make to the research field should be better highlighted in the last paragraph.

We have restructured the Introduction as suggested by Reviewer 1. We have started with the physical mechanisms that govern the monsoon atmospheric circulation (referring to updated

Fig. 1) and the need to improve the basic seasonality of the monsoon in coupled GCMs. This is then followed by the overlooked effect of chlorophyll on the South Asian monsoon. Then the physical and biological properties of the BoB. The last paragraph better highlights the contribution of our study to the research field (please see specific comment 1 above).

**Lines 101-102: I believe Wetzel et al. (2006) used a coupled GCM, whereas your study actually does not. This statement is therefore incorrect.**

Reviewer 1 is correct about Wetzel et al. (2006). The statement has been corrected to:

"This study analyses the direct effect of BoB seasonally varying chlorophyll concentrations on the South Asian monsoon in an atmospheric GCM that is coupled to a mixed layer thermodynamic ocean model."

Lines 127-128: I suggest adding a more expanded definition of h1 and h2, for instance: the e-folding depth, or attenuation depth, is the depth where surface radiation decreases by 1/e (or 37%) of its initial surface value.

An expanded definition of scale depth has been provided as follows:

" $h_1$  and  $h_2$  are the scale depths of red and blue light, respectively. The scale depth, or *e*-folding depth, is defined as the depth where solar irradiance is approximately 63% less than its surface value  $(1 - e^{-1})$ ."

Lines 138-139: Please state already here which Jerlov water type is used in the simulation with constant attenuation depth.

We have now stated that the value of  $h_2$  from Jerlov water type IB was used to represent the global average solar penetration depth.

"In this study,  $h_2$  of 17 m from Jerlov water type IB is used to represent the global average solar penetration depth in MC-KPP."

Lines 141-145: How does the relaxation of ocean temperature to observations affect your results? Which is the relaxation time scale and how sensitive are your results to the relaxation time scale?

There is no relaxation of ocean temperature or salinity to observations in the simulations we analyse, only fixed (prescribed) seasonal cycles of temperature and salinity corrections that are computed as tendency terms. These ocean tendency terms remain the same for the perturbation run, and thus the ocean temperature and salinity differences between the control and perturbation runs are entirely due to the perturbation itself and subsequent atmospheric feedbacks.

These temperature and salinity tendency terms are computed by running a separate 10-year MetUM-GOML relaxation simulation, where temperature and salinity profiles are constrained to the reference climatology with a relaxation timescale of 15 days. Prescribing the mean seasonal cycle of temperature and salinity tendency terms from the relaxation simulation into the 30-year control and perturbation simulations (1) represents mean oceanic advection; (2) corrects for biases in atmospheric surface fluxes. Hirons et al. (2015) found that a relaxation timescale of 15 days in the relaxation simulation is optimal to minimise temperature biases in the free-running MetUM-GOML simulation. Lengthening the timescale reduces the amplitude, and hence the efficacy, of the corrections; shortening the timescale overly constrains the

relaxation simulation, such that the mean state does not resemble the mean state in the freerunning simulation, which makes the tendency terms ineffective (Hirons et al., 2015). We have provided more explanation about the choice in relaxation timescales.

"The ocean temperature and salinity correction method of Hirons et al. (2015) is used to constrain the MC-KPP mean state to account for missing advection and biases in atmospheric surface fluxes. The method uses a separate 10-year MetUM-GOML relaxation simulation in which temperature and salinity are relaxed with a 15-day timescale to an observed seasonal cycle, here the 1980-2009 climatology of Smith and Murphy (2007). A relaxation timescale of 15 days is optimal to produce temperature and salinity tendency terms that minimise biases in the free-running simulations (Hirons et al., 2015)."

**Lines 155 and later line 181: why say "h2 values (i.e. chlorophyll concentration)"? It is clearly not the same quantity.**

This statement was used to reinforce the idea that prescribed  $h_2$  values in MC-KPP represent chlorophyll concentrations. We have removed the statement " $h_2$  values (i.e. chlorophyll concentration)" from line 155, as it was clearly stated at the beginning of the paragraph that  $h_2$ is representative of chlorophyll concentration. On line 181, we have replaced " $h_2$  values (chlorophyll concentration)" with " $h_2$  values (representative of chlorophyll concentration)" to remind the reader what  $h_2$  represents.

*"...with differing prescribed h2 (representative of chlorophyll concentration):..."*

**Line 173: Fig. 1 could show in gray shading the grid points in which chl-a was not determined by satellite.**

Thank you for the good suggestion. The location of undetermined chlorophyll concentrations from satellite has now been shaded in grey and the missing  $h_2$  model grid points in MC-KPP are now shown by the black hatching. Undetermined chlorophyll concentrations shown by the grey shading in Fig. 1 are mentioned.

"Hence, remotely sensed chlorophyll-a concentrations were not determined in the eutrophic coastal regions of the Ganges and Irrawaddy river deltas because of the large amount of suspended organic and terrestrial material (grey shading; Fig. 1; Tilstone et al., 2011)."

The missing  $h_2$  data in MC-KPP shown in Fig. 1 is now also mentioned.

"The missing  $h_2$  data in MC-KPP were typically associated with coastal regions where the land fraction was between 1 and 0, which includes the narrow isthmus of Thailand and the low-lying land of the Ganges delta (black hatching; Fig. 1)."

**Lines 184-185: 30-years of simulation sounds quite short, also considering that computational costs are not very high (line 149). Is the model in an adjusted state?**

After the 10-year relaxation simulation and computation of temperature and salinity corrections, the two 30-year free-running simulations require a small spin-up time (1 year) as the ocean is by then adjusted to a mean state. A 30-year simulation is an adequate climate-length simulation that has demonstrated a clear chlorophyll-induced signal over the BoB. The low computational costs are judged relative to a simulation with a fully dynamic ocean. At the 90 km resolution used here, MetUM-GOML still requires considerable supercomputing

resources that limits the length of simulation we can perform. For reference, MetUM with a dynamical ocean costs approximately 30% more per model year than MetUM-GOML.

"A mean seasonal cycle of daily temperature and salinity tendencies that is computed from this relaxation simulation is applied to the 30-year chlorophyll perturbation simulations. The spin-up time is small (1 year) as the ocean is adjusted to a mean state after the relaxation simulation."

Lines 200-212: I think these two paragraphs should go in Section 2.2. Indeed, at the first sentence of Section 2.2 I was already wondering how you compute h2 based on chlorophyll data.

We respectfully disagree with Reviewer 1 about moving the two paragraphs (lines 200-212) from Section 2.3 to 2.2. Section 2.2 focuses on the satellite chlorophyll-a data used, such as how the satellite product is derived and reasons why chlorophyll-a is undetermined in turbid coastal waters. Section 2.3 presents the experiment set-up in chronological order starting with the experiment design, then the explanation of the BoB domain and regridding scheme, then  $h_2$  conversion and finally the method to interpolate missing  $h_2$  values. Moving the paragraphs would remove the methodological order and possibly further confuse the reader.

Lines 213-226: This section doesn't read well, as it contains three very different sets of information (statistical significance computation, VIMF computation, regridding of observed precipitation). Please restructure. Part of the information could maybe go in the figure captions. We have attempted to improve the organisation and structure of this section. We have moved the statistical significance sentence (Line 213) to Fig. 4 caption (see end of document). A paragraph break was added on line 222 to separate VIMF computation and TRMM regridding description. Sentences have been rearranged to improve the description of the regridded TRMM product.

"Vertically integrated moisture fluxes (VIMF) were used to evaluate the water vapour transport sourced from the chlorophyll-forced BoB to the surrounding Indian subcontinent. The VIMF was calculated as

$$VIMF = \frac{1}{a} \int \vec{u} q \, dp$$

(2)

(3)

where  $\vec{u}$  is the horizontal wind velocity, q is the specific humidity, g is the acceleration due to gravity, p is pressure and the integration was between 1000 and 100 hPa. Note that  $\vec{u}q$  was output directly by the model as monthly mean values. VIMF divergence was used to evaluate the precipitation rate changes that are due to changes in water vapour divergence. The VIMF divergence was calculated as

$$VIMFD = \frac{1}{a} \int \frac{\partial \vec{u} q}{\partial x} + \frac{\partial \vec{v} q}{\partial y} dp$$

where the integration was between 1000 and 100 hPa.

"The observed monthly 18-year (1998–2015) climatological precipitation rate measured from the Tropical Rainfall Measuring Mission (TRMM) 3B42 satellite product (Huffman et al., 2007) was used to diagnose the bias in the model precipitation rate. An area-weighted re-gridding scheme was used to reduce the 0.25° horizontal resolution of the observed monthly climatological precipitation rate to match the horizontal resolution of MetUM-GOML."

**Line 245: Unless I'm mistaken, Fig. 6 is cited before Fig. 5.**

Reviewer 1 is correct, and we have changed the Figure numbers to ensure that Fig. 5 is cited before Fig. 6.

Lines 251-252: please explain the mechanism better: despite the SST increase, evaporation decreases because air humidity increases?

We have improved the explanation of the mechanism counteracting an increase in SST.

"The upward latent heat flux increases (Fig. 4k and 4l) due to an increase in the specific humidity at the surface, which is associated with the higher SST. This increase in SST is therefore offset by the negative feedback from the latent heat flux increase."

Line 255 and 265: "caused by" instead of "corresponds"? We have replaced "corresponds" with "caused by".

Lines 255-259 (same for 289-291): I suggest not mixing mechanism understanding and bias improvement in one paragraph. Indeed, maybe all the part on bias improvement could be shifted to the discussion?

We have moved the bias improvement to the Discussion and Conclusions section. As Fig. 7 is mentioned later in the manuscript, we have now re-numbered it to Fig. 13 (see end of document).

Lines 263-272: This paragraph is somewhat cumbersome. I suggest condensing the relevant information in fewer sentences.

We have condensed and restructured this paragraph. Please note Fig. 8 has been renumbered to Fig. 7.

"The values of  $h_2$  continue to decrease in the BoB open ocean into July and August (Fig. 7a and 7b), as high chlorophyll concentrations off the continental shelf and SMC encroach further into the central BoB. The lowest monthly average  $h_2$  in the SMC region and the central BoB occurs in August with a value of 12 m and 15 m, respectively (Fig. 7b). During September and October the SMC weakens and the observed high chlorophyll concentrations retreat back to the coast, increasing average  $h_2$  in the SMC region and the central BoB to around 16 m (Fig. 7c and 7d). Meanwhile, along the northwest BoB during October, monthly average  $h_2$  decreases to 13 m, as observed high chlorophyll concentrations retreat back onto the continental shelf (Fig. 7d)."

Line 274: co-located?

We have replaced "collocated" with "co-located".

Line 274: no comma after whereas?

We have removed the comma after "whereas".

**Line 275: Increased SST also speeds up evaporation.**

We have added the increase in SST as another factor in increasing latent heat flux.

"In July, an increase in SST and a slight increase in alongshore windspeed over the west BoB increases the upward latent heat flux (Fig. 7m), but this does not significantly change precipitation rate (Fig. 7q)."

Lines 300-306: This part seems more suitable for the Discussion section.

We have moved lines 300-306 into the Discussion and Conclusions section. We have slightly expanded the sentence on line 300 to include the effect of the Silk Road pattern on air temperature and rainfall anomalies in east Asia, as stated by Ding and Wang (2005).

"This indirect remote response resembles the effect of the "Silk Road" pattern; a stationary Eurasian-Pacific Rossby wave train that occurs during the Northern Hemisphere summer and produces significant air temperature and rainfall anomalies in east Asia (Ding and Wang, 2005)."

Section 3.3 (lines 309-371): if I understand correctly, Eq. 4 is used to estimate the relative contribution of attenuation depth, incoming shortwave radiation and MLD changes on the radiative heating. Throughout the section, you then refer to these results only in the text. I think it would be more useful to see the relative contribution of these processes in a separate figure. For instance, in Fig. 11 you could show, instead of the Qsw and MLD changes, the relative contribution that Qsw and MLD have on the RHR.

We agree with Reviewer 1 that it would be useful to the show the relative contributions of  $Q_{sw}$ , MLD and  $h_2$  on the RHR. Please see new Fig. 11 at end of the document.

**Line 320: I'm acquainted with cp=3850 J/(kg\*K). Please clarify.**

We erroneously used the  $c_p$  value for freshwater. We have now changed this to the  $c_p$  value of seawater (3850 J kg-1 K-1). Figures and values of calculated radiant heating rates have been corrected, but this relatively minor change does not affect our conclusions.

Line 324: "in" too much.

We have removed unnecessary instances of "in".

Lines 329-330: please be more explicit on why you selected these two regions. We have been more explicit as to why we have selected the SMC and Irrawaddy Delta regions.

"The two regions are an important source of heat and moisture for the June and October precipitation rate perturbations and display distinctive chlorophyll regimes. The SMC is an open ocean region that displays large seasonal changes in  $h_2$ , whilst the Irrawaddy Delta is a coastal region that displays continuously low  $h_2$ ."

Lines 356-370: please state more explicitly why a deeper MLD leads to smaller biologicallydriven changes in SST. If I understand correctly, the mechanism you are referring to is: an increased radiative heating in the upper levels caused by biology leads to radiative cooling in the layers below. A MLD deeper than the attenuation depth causes this dipole to mix, and therefore (assuming constant Qsw) the net effect of biological heating on ML temperature should be null. To that one should also add the fact that higher surface SSTs lead to increased ocean heat losses by evaporation. Therefore, on annual average the areas of high biology (and deep MLD) should actually experience surface cooling. I wonder whether you see this effect in your model?

Reviewer 1 is correct about the mechanism. We have stated why a deeper MLD reduces the effect of biological heating in a separate sentence.

"When the MLD deepens below 10 m, the biological-induced effects of the increased radiant heating rates above 10 m and reduced radiant heating rates below 10 m are mixed, reducing the net effect of biological heating on mixed layer temperature. In

June, the MLD deepens to 16 m (Fig. 12a), resulting in a smaller average radiant heating rate change of  $0.4 \,^{\circ}$  month-1 (Fig. 12e)."

What the reviewer has highlighted at the end of their comment is the first link in a complex chain of events and possible feedbacks through changes in latent heat flux. However, please note that if the SST is not increased due to a deep MLD, then there would be no local reason for enhanced latent heat fluxes, thus we would not expect negative SST anomalies. The SST and latent heat flux respond rapidly to dynamical feedbacks from atmospheric circulation changes that are induced by biological warming both locally and remotely.

Lines 356-371: It is intriguing to see that the presence of biology and associated radiative heating do not modify the MLD. Could you comment on that?

We have added sentences about the MLD not responding to changes associated with the radiant heating in the SMC and Irrawaddy Delta regions.

"There is also no change in MLD in response to reduced  $h_2$  in the perturbation run. The increase in local wind speed of 0.8 m s-1 is likely to have de-stratifying effects on the upper ocean that oppose the stratifying effects of increased mixed layer radiant heating."

"As in the Irrawaddy Delta region, there is no change in MLD in response to biological warming in the SMC region due to an increase in local wind speed of 0.8 m s-1, which is likely to oppose the stratifying effects of increased mixed layer radiant heating."

Discussion-Conclusions: The authors could consider joining the Discussion and the Conclusions. My suggestion: first start with a paragraph summarizing the main results (as in lines 435-455), then continue with the discussion points, and finally conclude with the open questions and outlook. As it is now, some parts of the Conclusions seem more apt to be in the Discussion.

We agree that some parts of the conclusion are more suitable for the discussion. We therefore adopted the suggestion to combine the Discussion and Conclusion sections together. The summary of the main results from the Conclusions section has been moved to the beginning of the Discussion and Conclusion section, followed by a paragraph on the Silk Road pattern moved from Section 3.2. This is then followed by a new paragraph explaining the schematic that summarises the effect of chlorophyll-induced warming on monsoon precipitation (See Fig. 14 at end of document).

Lines 388-389: The MLD changes of 1 m cannot readily be seen in Fig. 11. It would be more useful to show spatially averaged MLD anomalies as suggested in point 3 of my specific comments.

As mentioned in the third specific comment, we have reduced the size of the domain and removed political boundaries to improve the extraction of relevant information from Figs. 10 and 11.

Lines 390-391: This is an interesting statement, which would imply that in coastal regions the local effect of high chlorophyll dominates the perturbation minus control anomalies in stratification, whereas in the open ocean remote effects through wind speed changes play a leading role. Some more analysis should be shown to substantiate this statement.

Upon further analysis we have identified that the stratifying effect of biological warming occurs in localised regions where  $h_2$  is very low (< 8 m) and where there is little or no change

in windspeed. Therefore, windspeed changes still dominate upper ocean stratification and airsea exchange in both coastal and open ocean regions. We have corrected line 390 below.

"The effect of high chlorophyll concentrations in these localised coastal regions appears to have altered upper-ocean thermal stratification when there is little or no change in windspeed, while in the majority of the BoB, changes in windspeed primarily alter upper-ocean thermal stratification."

Lines 423-425: The discussion on the effect of subseasonal chlorophyll variability on the Boreal Summer Intraseasonal Oscillation comes somewhat out of the blue, since you have not mentioned any form of intraseasonal or interannual variability in the paper. Please introduce this paragraph better to put the reader in the picture.

We have removed this paragraph from the manuscript as it diverges from the main focus of the study, which is the effect of chlorophyll on the seasonality of the monsoon.

Lines 430-431: Which feedbacks and processes are you missing by using an ocean mixed layer model instead of an ocean GCM (and by using an imposed chlorophyll distribution instead of an interactive biogeochemical model)? Do you expect the results from your modeling setting to be rather overestimating or underestimating the "true" response?

By using an ocean mixed layer model instead of an ocean GCM we do not simulate any dynamical changes to the modified vertical profile of solar absorption, which might further influence SST and thus create feedbacks with the atmosphere. By not simulating the chlorophyll in an interactive biogeochemical model we miss any potential biological feedbacks between solar absorption, mixed-layer depth, nutrient availability and the depth of the chlorophyll maximum. However, these feedbacks would be complex and vary in both space and time, with subsequent feedbacks. Therefore, it would not be appropriate to speculate on the "true" response without conducting such experiments, which would be beyond the scope of this study. Nevertheless, by including the observed chlorophyll variability and simulating the thermodynamic air-sea interaction and its subsequent impact on monsoon rainfall, we believe we capture the most important processes that are missed by coupled climate models with fixed values for solar absorption.

We have discussed the feedbacks and processes that are missing from MC-KPP in a new separate paragraph:

"The limitations of representing ocean dynamics as a mean seasonal cycle means that MC-KPP cannot capture any ocean dynamical response to biologically-induced changes to ocean properties (e.g., changes to ocean temperature and salinity transports). Previous studies have shown large effects of chlorophyll on ocean dynamics in the equatorial Pacific (e.g., Nakamoto et al., 2001; Murtugudde et al., 2002) and in mid- to high-latitude regions (e.g., Manizza et al., 2005; Patara et al., 2012). Modified biological warming at the surface or perhaps modified solar radiation penetration below the mixed layer could affect the dynamics of SMC and SLD in the BoB. Missing modes of variability in MetUM-GOML that rely on a dynamical ocean, such as ENSO and IOD, could combine non-linearly with the ocean anomalies induced by biological warming, with implications for monsoon rainfall. Further research using a fully dynamical coupled ocean-atmosphere GCM is required to show the dynamical changes and feedbacks of biological warming on the BoB oceanic and atmospheric system."

Line 459 onwards already outlines the missing biological feedbacks of an imposed chlorophyll distribution and how a coupled interactive biogeochemisty model might represent those feedbacks. We have added a sentence to line 458 to acknowledge that an imposed chlorophyll distribution in MC-KPP leads to missing biological feedbacks.

"The imposed seasonally and spatially varying  $h_2$  in MC-KPP eliminates any biological response to secondary feedbacks in the ocean. A coupled biogeochemistry model linked to an ocean-atmosphere GCM would be needed to further understand secondary feedbacks on phytoplankton productivity."

Line 436: Please be more specific on the model you used. (Coupled ocean-atmosphere GCM is misleading).

This sentence has been removed.

Lines 450-454: The inclusion of biological heating reduces the precipitation biases in some locations, which is good, but does it increase the biases elsewhere? (e.g. over Myanmar?) The model bias worsens in central Myanmar during June by approximately 2 mm day-1, however, the increase in precipitation rate in the perturbation run is not significant and therefore unlikely to be directly or indirectly caused by changes in biological warming. We therefore don't draw attention to this in the text.

**New References**

Li, C., and Yanai, M.: The onset and interannual variability of the Asian summer monsoon in relation to land-sea thermal contrast, J. Clim., 9, 358-375, https://doi.org/10.1175/1520-0442(1996)009<0358:TOAIVO>2.0.CO;2, 1996.

Manizza, M., Quéré, C. L., Watson, A. J., and Buitenhuis, E. T.: Bio-optical feedbacks among phytoplankton, upper ocean physics and sea-ice in a global model, Geophys. Res. Lett., 32, L05603, http://doi.wiley.com/10.1029/ 2004GL020778, 2005.

Nakamoto, S., Kumar, S. P., Oberhuber, J. M., Ishizaka, J., Muneyama, K., and Frouin, R.: Response of the equatorial Pacific to chlorophyll pigment in a mixed layer isopycnal ocean general circulation model, Geophys. Res. Lett., 28, 2021-2024, http://doi.wiley.com/10.1029/2000GL012494, 2001.

Patara, L., Vichi, M., Masina, S., Fogli, P. G., and Manzini, E.: Global response to solar radiation absorbed by phytoplankton in a coupled climate model, Clim. Dyn., 39, 1951-1968, http://link.springer.com/10.1007/s00382-012-1300-9, 2012.

Webster, P. J., Magaña, V. O., Palmer, T. N., Shukla, J., Tomas, R. A., Yanai, M., and Yasunari, T.: Monsoons: Processes, predictability, and the prospects for prediction, J. Geophys. Res. Ocean, 103, 14451-14510, http://doi.wiley.com/10.1029/97JC02719, 1998.

---

## Author Comment (AC2) · 15 Sep 2020

**Author Comments 2 on "The effect of seasonally and spatially varying chlorophyll on Bay of Bengal surface ocean properties and the South Asian Monsoon"**

We would like to thank Reviewer 2 who provided constructive and insightful comments that have greatly improved the revised manuscript. We have incorporated all their suggestions where possible. Reviewer 2 comments have been reproduced in black with authors response in blue and excerpts from the revised manuscript in italics. The revised and renumbered figure is included at the end of the document.

**Response to Reviewer 2**

1. It would be good to quantify the impact of chlorophyll on the changes in SST. Maybe the Abstract or the Conclusion could capture this important aspect.

We have now provided the change in SST due to biological warming in the Abstract as suggested.

"The largest SST response of  $0.5 \,^{\circ}$ C to chlorophyll forcing occurs in coastal regions, where chlorophyll concentrations are high (> 1 mg m-3), and when climatological mixed layer depths shoal during the intermonsoon periods."

2. Could the authors clarify the errors in their estimates of impact, given that the horizontal resolution of the effective coupled model is actually governed by the 90 km grid spacing of the MetUM-GOML3.0? I understand that the 4km satellite observations have been re-gridded to the 90km model grid. If this is not correct, could the authors specify the horizontal resolution of the ocean model (which is really okay for the vertical – with 100 levels in the upper 1000 m, of which 70 levels are in the top 300m)?

Regridding the 4 km satellite observations to the coarser 90 km model grid has implications on the representation and impact of biological warming in the BoB, which has been discussed in comment 8 below. Running the ocean model at a finer resolution than the atmosphere (90 km) would have little value, because the ocean properties (e.g., SST) must be averaged to the atmospheric grid before they are passed back to the atmosphere. Similarly, the ocean receives surface heat, moisture and momentum fluxes on the 90 km atmospheric grid, so these fluxes would be identical across the atmospheric gridbox even if the ocean were run at higher resolution. The only value from a higher-resolution ocean would come from non-linear effects of sub-90 km spatial variability in chlorophyll on the ocean properties (i.e., if the SST averaged across a 90 km x 90 km of finer-resolution ocean columns with higher-resolution chlorophyll differed from the SST of single 90 km x 90 km gridbox with averaged chlorophyll concentrations). We are unable to speculate about the errors in using a coarser-resolution model with unresolved mesoscale chlorophyll distributions. A separate study using a high-resolution fully-coupled model is needed to further investigate the mesoscale impact of chlorophyll on ocean properties.

Reviewer 2 is correct about the 4 km satellite observations that were re-gridded to the  $\sim$ 90 km oceanic and atmospheric model grid. We have now added more detail about the horizontal resolution of MC-KPP and MetUM in Section 2.1.

"The atmospheric and oceanic horizontal resolution is N216 (0.83° longitude x 0.56° latitude), which corresponds to a horizontal grid spacing of approximately 90 km."

"MC-KPP consists of a grid of independent one-dimensional columns, with one column positioned under each atmospheric grid point at the same horizontal grid spacing as MetUM GA7.0."

3. Given the 90 km horizontal resolution of the ocean model, how reliable are the inferences for the "coastal chlorophyll impacting the SST" results? Is there really a coast in the ocean model? Should we interpret them as near-coastal? The whole BOB would be a 25 x 25 grid ocean at 90 km resolution. Most of the coastal regions are resolved by such a grid with less than 2 grid points. Maybe the authors could clarify this with care, so that future studies can build upon this limitation.

We have shown that coastal chlorophyll influencing SST is a robust result. However, a higherresolution model would likely focus this response in a narrower region close to the coast. Future study using a high-resolution model is needed to further investigate the impact of coastal chlorophyll. See comment 8 below about the limitations of the horizontal resolution of MC-KPP presented as a new paragraph.

To clarify to Reviewer 2, there is no immediate transition from land to sea in MC-KPP. Instead, the coastal region in MC-KPP is represented over multiple grid points with points that are partially ocean and partially land. At these points, surface properties (e.g., fluxes, temperatures) are computed separately for the land and sea portions of the points. At these points, MC-KPP receives the "ocean part" of the fluxes from the atmospheric model; the atmospheric model combines the SST from MC-KPP with the surface temperature from the land model to construct a gridpoint-mean temperature. We have provided a clearer definition of the coastal region in MC-KPP in Section 2.1.

"The coastal region in MC-KPP is represented with columns that are partially ocean and partially land. The surface properties for ocean and land are computed separately in MC-KPP and the mean grid point temperatures are computed in the atmospheric model by combing the ocean and land surface temperatures from MC-KPP."

4. Furthermore, as authors point out in lines 205-212, there are lots of missing values of h2 in coastal regions (not surprisingly) – this again could influence the inferences in lines 384-400. Please also see comment 7 below.

The relatively few missing  $h_2$  values in the coastal region are due to the highly turbid coastal water close to the coast that is excluded in the chlorophyll satellite product. We have added black hatching to Fig. 1 to show missing  $h_2$  grid points in MC-KPP (new figure included at the end of this document). We have added a paragraph discussing the limitations and implications of the interpolated  $h_2$  values. Please see comment 7 below.

5. The first paragraphs of Section 3.1 and 4.1 are hard to follow. This reader was getting very confused with the increase/decrease and high/low ordering of sentences. Maybe talk about changes focusing on regions first and then the months. A minor issue is the frequent usage of the phrase 'in reality' in both paragraphs. This phrase occurs five times in lines (233-239) and another five times in lines (264-271) – almost once in every sentence. Maybe use 'observed', 'satellite' etc.

We have restructured the first paragraph of Section 3.1 and removed unnecessary instances of "in reality", replacing them with 'observed' as suggested.

"The BoB surface ocean responds to the imposed annual cycle of h2 in the perturbation run during the onset of the southwest monsoon. In the central BoB, values of h2 increase above the global constant of 17 m, as observed surface chlorophyll concentrations are low during southwest monsoon onset (Fig. 4a-c). Along the northern BoB coast, values of h2 are as low as 5 m, as observed surface chlorophyll concentrations in coastal areas are higher than those in the central BoB (Fig. 4a-c). During May and June, the values of h2 decrease and mixed-layer solar absorption increases in the northwest BoB, as observed high coastal chlorophyll concentrations extend oceanward across the continental shelf (Fig. 4b-c). In the southwest BoB, the imposed h2 decreases in May and June to 14 m, as the strengthening SMC advects high chlorophyll concentrations from the south coast of India and Sri Lanka (Fig. 4b-c). "

We have replaced the months with "spring" and "autumn" intermonsoon, and restructured sentences in the first paragraph of Section 4.1.

"During the spring intermonsoon, a peak in surface chlorophyll concentrations and shallow MLDs led to an increase in SST. During the autumn intermonsoon, another peak in surface chlorophyll concentration led to a similar, but weaker increase in SST due to deeper MLDs and stronger turbulent surface fluxes."

We have also condensed and restructured the first paragraph of Section 3.2 (see author comment from Reviewer 1 on lines 263-272).

6. Lines 245, 257 – the superscripts for units did not come through in my downloaded version. There are other similar occurrences throughout the paper. Please check.

You are right that the superscripts for the units failed to appear in the uploaded manuscript. Apologies for that. We will ensure that the superscripts are shown in the revised manuscript.

7. It is encouraging to note that the authors have used the Satellite-derived chlorophyll concentrations to h2 using a fifth-order polynomial parameterization. They reported the improvement of SST and precipitation in the coastal region. However, the coastal BoB is mostly dominated by river water, where the above algorithm (5th order polynomial to get h2) might not be totally applicable. The authors have actually interpolated and/or extrapolated to fill the data gaps (page 5, 205). This could possibly lead to a positive bias in precipitation in the head BoB in the post-monsoon period (Figure 8). Could the authors please clarify these two aspects or limitations.

It is a good suggestion to discuss the limitations in the chlorophyll satellite product and solar penetration depth parameterisation. Oceanic constituents such as CDOM and suspended sediments are falsely interpreted as a chlorophyll-a concentration, leading to overestimation of chlorophyll-a concentrations in coastal regions. We have now referred to the Ganges river delta as this is a region susceptible to inaccurate and undetermined chlorophyll and  $h_2$  due to the high levels of coastal turbidity. An overestimation in chlorophyll-a concentration could lead to an overestimate in biological warming with repercussions on BoB precipitation rates.

"The derivation of the imposed annual cycle of  $h_2$  in coastal regions has limitations. Firstly, the ocean colour algorithms used to determine chlorophyll concentrations from satellite are not completely effective in turbid coastal waters (Morel et al., 2007; Tilstone et al., 2013). Organic and inorganic constituents such as Coloured Dissolved Organic Matter (CDOM) and suspended sediments strongly attenuate blue light and are thus falsely identified as a chlorophyll-a pigment, which typically leads to an overestimation in chlorophyll concentration (Morel et al., 2007). Secondly, the Morel and Antione (1994) chlorophyll parameterisation is not applicable for coastal waters, as the parameterisation is based on the absorption by chlorophyll-a pigment and not by the attenuation of other in-water constituents. Missing h2 values in the Ganges river delta are interpolated from neighbouring h2 values that are likely associated with satellite product and parameterisation uncertainty. The Ganges coastal region has been found to influence spring intermonsoon SST and precipitation rates in the northern BoB. Possible positive biases in chlorophyll concentration in the Ganges river delta are likely to lead to an overestimation in the coastal biological warming, SST and precipitation rate increase. Ocean colour algorithms to determine proxy coastal chlorophyll concentrations are still an area of active research (Blondeau-Patissier et al., 2014). Future studies should consider the attenuation of solar radiation from other oceanic constituents in turbid coastal regions to better represent radiant heating in the upper ocean."

8. The BoB is a highly eddy-active region, which has a significant impact on the chlorophyll distribution and on the air-sea interaction (page 13, 570). The authors could expand on how finer resolution ocean models might be helpful in the future for resolving both eddy activities in the open region and mesoscale to sub-mesoscale features in coastal BOB.

We have now mentioned the use of high-resolution, fully dynamical models to improve the representation of mesoscale chlorophyll distributions and eddy activity that influences biological productivity.

"The mesoscale and sub-mesoscale spatial variability of h2 and associated oceanic processes is inadequately represented in MC-KPP due to its coarse horizontal resolution. The coastal region in MC-KPP is represented by multiple grid points that are partially ocean and partially land at an approximate 90 km horizontal resolution. Such a resolution means that at the coastlines, the mesoscale coastal chlorophyll concentration features and the corresponding solar penetration depths are poorly resolved. Future studies should consider using a high-resolution, fully dynamical model to accurately resolve the coastline and associated solar penetration depths. The simulated dynamics would improve the representation of mesoscale eddy activity along the coast and open ocean, which increases biological productivity (Kumar et al., 2007) that in turn increases local solar radiation absorption."

9. In the BoB (like in other oceans), Chlorophyll maxima is generally not at the surface level. It varies from 10 m to 80 m (Pramanik et al., 2020). The impact of the deep chlorophyll maxima and its relationship with the surface chlorophyll and SST variations may be explored in a separate future work, but worth a mention.

This is an interesting point that we agree should be mentioned close to the end of the Discussion and Conclusions.

"The chlorophyll concentration in the BoB upper ocean is not homogeneous with depth. In situ observations show that the vertical depth of chlorophyll maxima varies between 10 and 80 m (Thushara et al., 2019; Pramanik et al., 2020), often occurring at depths undetected by satellite radiometer sensors (Huisman et al., 2006). Variations in the vertical depth of the chlorophyll maxima would vary the vertical depth of enhanced radiant heating. However, if the depth of the chlorophyll maxima occurs at a depth where solar radiation is significantly reduced (e.g., at the euphotic depth where solar radiation is ~1% of its surface value), then the change in local radiant heating at that depth would be negligible (Morel and Antione, 1994). Indeed, observations show the occurrence of intense deep chlorophyll maxima in the BoB at depths of 20 to 40 m (Thushara et al., 2019), which might have a strong influence on local mixed layer radiant heating and vertical heat distributions. Hence, the effect of nonuniform chlorophyll concentration profiles on upper ocean radiant heating and SST requires further investigation."

**New References**

Blondeau-Patissier, D., Gower, J. F., Dekker, A. G., Phinn, S. R., and Brando, V. E.: A review of ocean color remote sensing methods and statistical techniques for the detection, mapping and analysis of phytoplankton blooms in coastal and open oceans, Progress in Oceanography, 123, 123-144, https://linkinghub.elsevier.com/retrieve/pii/S0079661114000020, 2014.

Huisman, J., Pham Thi, N. N., Karl, D. M., and Sommeijer, B.: Reduced mixing generates oscillations and chaos in the oceanic deep chlorophyll maximum, Nature, 439, 322–325, http://www.nature.com/articles/nature04245, 2006.

Pramanik, S., Sil, S., Gangopadhyay, A., Singh, M. K., and Behera, N.: Interannual variability of the chlorophyll-a concentration over Sri Lankan Dome in the Bay of Bengal, International Journal of Remote Sensing, 41, 1-18, https://www.tandfonline.com/doi/full/10.1080/01431161.2020.1727057, 2020.

Tilstone, G. H., Lotliker, A. A., Miller, P. I., Ashraf, P. M., Kumar, T. S., Suresh, T., Ragavan, B. R., and Menon, H. B.: Assessment of MODIS-Aqua chlorophyll-a algorithms in coastal and shelf waters of the eastern Arabian Sea, Cont. Shelf Res., 65, 14-26, https://linkinghub.elsevier.com/retrieve/pii/S0278434313002045, 2013.